# MADE: Benchmark Environments for Closed-Loop Materials Discovery

**Shreshth A. Malik** [1 2]   **Tiarnan Doherty** [2]   **Panagiotis Tigas** [2]   **Muhammed Razzak** [2]
**Stephen J. Roberts** [3]   **Aron Walsh** [4]   **Yarin Gal** [1 2]

## Abstract

Existing benchmarks for computational materials discovery primarily evaluate static predictive tasks or isolated computational sub-tasks. While valuable, these evaluations neglect the inherently iterative and adaptive nature of scientific discovery. We introduce **MA**terials **D**iscovery **E**nvironments (**MADE**), a novel framework for benchmarking end-to-end autonomous materials discovery pipelines. MADE simulates closed-loop discovery campaigns in which an agent or algorithm proposes, evaluates, and refines candidate materials under a constrained oracle budget, capturing the sequential and resource-limited nature of real discovery workflows. We formalize discovery as a search for thermodynamically stable compounds relative to a given convex hull, and evaluate efficacy and efficiency via comparison to baseline algorithms. The framework is flexible; users can compose discovery agents from interchangeable components such as generative models, filters, and planners, enabling the study of arbitrary workflows ranging from fixed pipelines to agentic systems. We demonstrate this by conducting systematic experiments across a diverse range of systems and algorithms, finding that adaptive planning becomes more important to discovery efficiency as the search space scales.

## 1. Introduction

Scientific discovery inherently runs in a closed loop. Researchers propose hypotheses, run experiments or simulations, and refine their ideas based on the outcomes (Popper, 2005). Failures can be as informative as successes, and strategies adapt as new evidence emerges. Materials discovery is no exception; promising candidates are proposed, evaluated, and iteratively refined, often through cycles of exploration, dead ends, and serendipitous insights.

Yet most computational benchmarks for materials discovery assume a one-way process (Figure 1a). Predictive benchmarks evaluate accuracy of predicting properties such as band gap, energy, and forces using fixed datasets (Dunn et al., 2020; Riebesell et al., 2025; Rubungo et al., 2025; Kirklin et al., 2015). Generative model evaluations on the other hand typically measure metrics such as stability, uniqueness and novelty for one-shot batch generation of candidates (Zeni et al., 2025; Merchant et al., 2023; Betala et al., 2025; Xie et al., 2022). These are valuable assessments of sub-components of pipelines, yet they evaluate models in isolation, divorced from the overall discovery workflow. In practice, researchers combine predictive models, generative models, filters, and heuristics in multi-stage pipelines, with the ultimate objective being the efficient experimental discovery of new materials.

In realistic discovery settings, evaluations such as a high-fidelity simulation or physical experiment can be very costly. Thus the efficiency of discovery algorithms under a limited query budget is important. Methods such as Bayesian optimization and active learning (Settles, 2012; Garnett, 2023) provide principled frameworks for adaptive experimentation and have shown strong performance in molecular and materials applications (Lookman et al., 2019; Rohr et al., 2020). These approaches differ in their modeling and acquisition strategies, but are typically developed in settings focused on efficiently optimizing a small number of continuous design variables toward a single global objective. Materials discovery, by contrast, is a multi-minima seeking problem that aims to find diverse stable or metastable compounds within a vast, discrete and constrained chemical space.

Concurrently, LLM-based scientific agents have demonstrated increasing capability at orchestrating multi-step workflows, integrating prior knowledge, and adapting strategies given feedback (Lu et al., 2024; Jia et al., 2024; Guo et al., 2025; Novikov et al., 2025; Abhyankar et al., 2026). While some recent benchmarks begin to probe aspects of scientific reasoning and tool use (Wang et al., 2025; Mirza et al.,

---
[1]OATML, Department of Computer Science, University of Oxford [2]Diffractive Labs [3]Machine Learning Research Group, Department of Engineering Science, University of Oxford [4]Thomas Young Centre and Department of Materials, Imperial College London. Correspondence to: Shreshth A. Malik <shreshth@robots.ox.ac.uk>.

*Proceedings of the 43$^{rd}$ International Conference on Machine Learning*, Seoul, South Korea. PMLR 306, 2026. Copyright 2026 by the author(s).

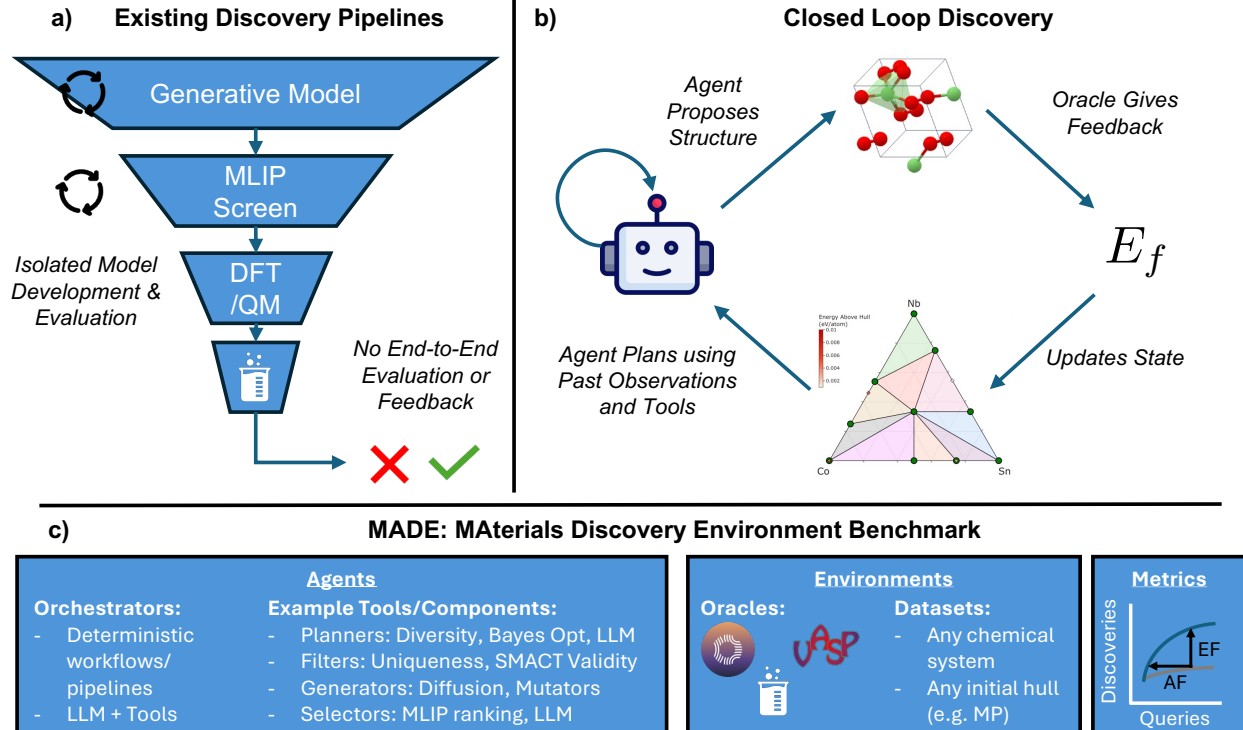

*Figure 1.* Conceptual overview of the MADE benchmark. **a)** Existing discovery pipelines and benchmarks follow a static filtering process, moving sequentially from generative models to increasingly expensive evaluation methods, without end-to-end feedback. **b)** MADE simulates a closed-loop discovery environment where agents iteratively propose candidates, receive oracle feedback (formation energy), and update their strategy. **c)** Modular, extensible components of the benchmark environments.

2024), there remains limited consensus on how to systematically evaluate agentic systems in open-ended, feedback-driven discovery settings (Song et al., 2025). Materials discovery provides a natural testbed for such systems: candidate proposals can be computationally verified in a closed loop, and there exists a rich ecosystem of computational tools and prior literature. This can enable controlled evaluation of discovery-relevant agentic behaviors such as planning and adaptive decision making.

To address these gaps in evaluations, we introduce a family of **MA**terials **D**iscovery **E**nvironments (**MADE**) that enable the evaluation of closed-loop discovery pipelines. In MADE, an agent or algorithm sequentially proposes candidate structures, receives feedback from the environment, and adjusts its strategy to efficiently discover novel thermodynamically stable compounds under a limited query budget. MADE is intentionally modular and composable; users can combine arbitrary planners, generators, filters, and scorers into pipelines, or evaluate fully agentic systems that can utilize environmental feedback. All experiments are specified via configuration files, facilitating reproducible comparisons and systematic ablations across discovery strategies.

Crucially, MADE supports discovery-centric evaluation metrics (Delgado-Licona & Abolhasani, 2023; Adesiji et al.,

2025). We quantify how quickly a method discovers new materials with respect to a baseline search strategy, extending the discovery-acceleration paradigm beyond machine learning interatomic potential (MLIP) screening (Riebesell et al., 2025) to the full discovery loop. This enables answering questions such as: what is the performance gain of using a better generative model? Do surrogate model rankings meaningfully accelerate discovery? Are agentic LLM systems more efficient than traditional search algorithms? In summary, our contributions are as follows:

- We introduce MADE, a family of environments for benchmarking closed-loop computational materials discovery, enabling the first systematic evaluation of full pipelines on open-ended discovery metrics.

- We benchmark strategies ranging from random search with generative models to agentic systems, across multiple system complexities, ablating the contributions of different components in pipelines.

- We find that agentic systems and adaptive search algorithms become more important for discovery efficiency as chemical complexity and search spaces scale, and as surrogate models reduce in efficacy.

## 2. Related Work

**Computational Materials Benchmarks** Benchmarking has played a central role in computational materials science. Foundational datasets such as the Materials Project and OQMD (Kirklin et al., 2015; Dunn et al., 2020) enable large-scale supervised learning for tasks including formation energy, forces, and band-gap prediction. These benchmarks focus on static predictive accuracy on fixed datasets. Matbench Discovery (Riebesell et al., 2025) shifted focus towards discovery-oriented evaluation by measuring a model's ability to rank candidates for stability using MLIPs on held-out structures. While this is a step forward, it still remains a screening benchmark: models operate on a fixed candidate pool without closed-loop adaptation. In contrast, MADE evaluates the closed-loop process of deciding what to generate, what to filter, and what to evaluate next. Meanwhile, papers on generative modeling (Zeni et al., 2025; Merchant et al., 2023; Park et al., 2025; Xie et al., 2022) and recent benchmarks for these models (Betala et al., 2025) focus on evaluating average unconditional generation quality rather than discovery acceleration metrics.

**Agentic Systems and AI-Driven Scientific Workflows** Recent advances in agentic systems, including LLM-based scientific agents (Lu et al., 2024; Guo et al., 2025; Novikov et al., 2025), tool-using design assistants (Jia et al., 2024; Inizan et al., 2025), and specialized agents for materials workflows (Badrinarayanan et al., 2025; Rubungo et al., 2025), have highlighted the potential of systems capable of multi-step reasoning, tool orchestration, and iterative refinement. Related work has also explored LLMs within classical adaptive search paradigms, such as Bayesian optimization and bandit-style decision making (Liu et al., 2024; Nie et al., 2024). However, most existing benchmarks assess static reasoning or tool use (Wang et al., 2025; Mirza et al., 2024; Jimenez et al., 2023; Nathani et al., 2025; Zhang et al., 2025) rather than long-horizon, feedback-driven discovery. Concurrent work on science-oriented benchmarks (Song et al., 2025; Huang et al., 2025) have started to move toward hypothesis–experiment–observation loops, but remain focused on structured evaluation settings rather than end-to-end discovery, and while Abhyankar et al. (2026) apply LLMs directly to materials discovery, evaluation only compares to generative models rather than full pipelines.

**Active learning and Bayesian Optimization** Active learning, Bayesian optimization (BO), and related experimental design methods provide a principled framework for sequential decision making under uncertainty, enabling efficiency optimization under limited evaluation budgets (Settles, 2012; Garnett, 2023; Rainforth et al., 2024). These methods combine surrogate models with acquisition functions that balance exploration and exploitation, and have

been widely applied in materials science to optimize properties in low-dimensional design spaces such as mapping phase diagrams (Lookman et al., 2019; Kusne et al., 2020; Rohr et al., 2020; Wang et al., 2022; Novick et al., 2024). Unlike classical black-box optimization, which targets a single global optimum, materials discovery is inherently multimodal, seeking a diverse set of local minima corresponding to stable or metastable compounds for experimental verification. MADE enables integration and evaluation of BO strategies within broader discovery pipelines.

## 3. MADE: MAterials Discovery Environment

We first define criteria and motivation for the design of our benchmark. We then introduce MADE[1] , the proposed framework for evaluating materials discovery strategies.

### 3.1. Desiderata for Discovery Benchmarks

We argue that a benchmark for materials discovery should satisfy three general desiderata:

- **Evaluate closed-loop discovery**. It should directly measure how effectively an end-to-end closed-loop system finds new materials (out-of-distribution to a given known set) to enable ablation of components in the pipeline.

- **Reflect realistic search challenges in materials science**. It should reflect the discrete, structured but sparse, multi-minima landscape of materials discovery.

- **Be general, scalable, and method-agnostic**. It should support controlled experiments across chemical systems, search-space sizes, and fidelity levels while remaining implementation-agnostic.

We use these desiderata to motivate MADE's design while remaining independent of specific pipeline implementation choices. We compare MADE to existing discovery benchmarks against these criteria in Appendix A.

### 3.2. Problem Definition

In MADE, an agent or algorithm interacts with a structured chemical environment by proposing candidate materials, receiving oracle feedback, and adapting its strategy over time. The goal is to uncover new thermodynamically stable compounds efficiently under a constrained oracle budget.

Let $S$ denote the chemical search space, where each candidate $s \in S$ is defined by its chemical composition and crystal structure. We assume access to an oracle $O : S \to \mathbb{R}$ which returns the predicted formation energy per atom $E_s$.

---

[1] https://github.com/diffractivelabs/MADE

Let $B \in \mathbb{N}$ denote the oracle query budget, and define $H_0 \subset S$ as the initial set of known reference materials.

An agent is defined by its discovery policy $\pi$ that depends on the history of observed (structure, energy) pairs,

$$\pi : \{(s_i, E_i)\}_{i=1}^{t-1} \to S. \qquad (1)$$

At each iteration $t \leq B$, the agent selects the next candidate structure $s_t \sim \pi$, the oracle evaluates its energy, $E_t = O(s_t)$, and the candidate is added to the set of known materials. After updating $H_t = H_{t-1} \cup \{s_t\}$, the convex hull $\mathrm{CH}(H_t)$ is recomputed. For each candidate $s \in H_t$, we calculate its energy above the convex hull as: $\Delta_{\mathrm{hull}}(s, H_t) \in \mathbb{R}$. A material is considered thermodynamically stable if its energy lies on or below the convex hull,

$$S_{\mathrm{stable},t} = \{s \in H_t \mid \Delta_{\mathrm{hull}}(s, H_t) \leq \epsilon\}, \qquad (2)$$

where $\epsilon$ is a small stability threshold (Bartel, 2022). Algorithm 1 shows a rollout of one episode in MADE. Pseudocode for the relevant classes is given in Appendix B.

---

**Algorithm 1** MADE episode rollout

---

**Require:** Chemical search space $S$, initial materials $H_0$, policy $\pi$, oracle $O$, budget $B$, threshold $\epsilon$
1: Initialize known materials $H \leftarrow H_0$
2: Evaluate energies $E_s = O(s)$ for all $s \in H$
3: Construct convex hull $\mathrm{CH}(H)$
4: **for** $t = 1$ to $B$ **do**
5:    $s_t \leftarrow \pi(\{(s, E_s) : s \in H\})$
6:    $E_t \leftarrow O(s_t)$
7:    $H \leftarrow H \cup \{s_t\}$
8:    Update $\mathrm{CH}(H)$ and stable set $S_{\mathrm{stable}}$
9: **end for**
10: **Return:** $S_{\mathrm{stable}}$

---

The sequence of proposed materials by the strategy is defined as: $Q_\pi = \{s_1, s_2, \ldots, s_B\} \subset S$. The objective is to design a policy $\pi$ that maximizes the total number of new stable materials discovered after $B$ queries:

$$\max_\pi |Q_\pi \cap S_{\mathrm{stable},B}|. \qquad (3)$$

This formulation explicitly treats discovery as multi-minima search which encourages diversity compared to black-box optimization objectives (Abhyankar et al., 2026).

### 3.3. Evaluation Metrics

In this work we assume oracle evaluations dominate the cost of discovery, where the cost of each oracle evaluation greatly exceeds the cost of intermediary computation required to plan and propose the query (Rainforth et al., 2024). This is often the case as the oracle in real-world use-cases is either

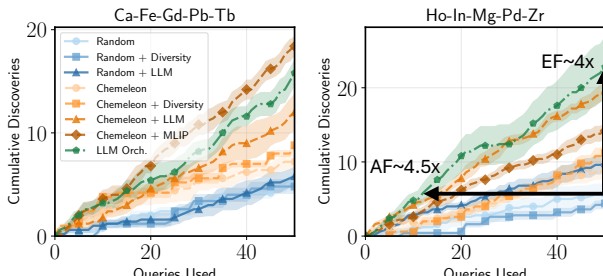

*Figure 2.* Example acquisition curves for different discovery policies on two quinary intermetallic systems. The acceleration factor (AF) and enhancement factor (EF) are shown for the LLM orchestrator policy with respect to a random generator baseline policy. Shaded regions are standard error in the mean across 5 episodes.

expensive DFT calculations or wet-lab experiments. We therefore emphasize discovery-centric metrics that explicitly account for sample efficiency.

**Independent metrics** These metrics evaluate a single policy without reference to a baseline.

- **mSUN** (Zeni et al., 2025; Betala et al., 2025). The fraction of *(meta)stable*, *unique* and *novel* materials proposed during an episode. A structure $s$ is counted if it is thermodynamically stable [Eq. (2)], not in $H_0$, and not among previously proposed structures. Structural novelty is enforced using `pymatgen.StructureMatcher`, which applies composition- and geometry-based similarity thresholds to prevent trivial perturbations from being counted as new discoveries.

- **Area Under the Discovery Curve (AUDC):** Let $D_\pi(t)$ denote the cumulative number of mSUN structures discovered by the policy after $t$ oracle queries. The AUDC is defined as $\mathrm{AUDC} = \frac{2}{B^2} \int_0^B D_\pi(t)dt$, where we normalize such that the maximum AUDC is 1. This captures both *how many* structures are discovered and *how efficiently* they are found.

**Relative metrics** To enable fair comparison across chemical systems of varying difficulty, we report metrics relative to a baseline strategy, $\pi_b$ (Figure 2). These relative metrics expose which strategies perform better on average within the same operational constraints, enabling principled algorithm selection across diverse discovery stacks.

- **Acceleration Factor (AF)** (Rohr et al., 2020): For a given number $k$ of discoveries, the acceleration factor is $\mathrm{AF}(k) = t_{\pi_b}(k)/t_\pi(k)$, where $t_\pi(k)$ is the number of oracle queries required by the policy to reach $k$ discoveries. AF quantifies how much more efficient a policy is compared to the baseline.

- **Enhancement Factor (EF)** (Rohr et al., 2020): For a given number $t$ of queries, the enhancement factor is $EF(t) = D_\pi(t)/D_{\pi_b}(t)$, measuring the multiplicative improvement in discoveries over the baseline.

**Additional and Extensible Metrics**   While the above metrics focus on discovery efficiency, MADE is easily extended to measure additional metrics, particularly in multi-objective settings. In experiments in this work, for example, we report composition and structural diversity metrics.

### 3.4. Environment

The environment is defined by the chemical system, initial known materials, and an oracle that evaluates proposals.

**Chemical Systems and Initial Materials**   The chemical space for exploration $S$ is defined by the constituent elements that make up the material. This allows for adjustable difficulty by varying system complexity (e.g. easy: binary metal oxides, medium: ternary intermetallic compounds, hard: quaternary and beyond), and stoichiometry bounds (maximum number of atoms in the unit cell). Initial known structures ($H_0$) can be retrieved from existing datasets (Horton et al., 2025; Barroso-Luque et al., 2024). By varying the size and composition of $H_0$, MADE can simulate settings ranging from well-explored chemical spaces to data-scarce regimes. New datasets can readily be incorporated as they emerge (Siron et al., 2025). To make evaluation of discovery policies fair, the same $H_0$ should be used for training components of discovery policies (Section 3.5) to prevent simple recall of known materials not in $H_0$.

**Oracles**   For efficient benchmarking and large-scale experimentation, MADE supports MLIP energy oracles, which offer fast approximate evaluations. Although MLIP evaluations are relatively inexpensive, they provide a realistic setting for studying sequential decision making and strategy adaptation. Crucially, MADE abstracts the oracle interface, allowing substitution with higher-fidelity evaluations such as density functional theory (DFT) calculations or experimental validation for simulation of realistic discovery campaigns.

### 3.5. Example Discovery Policies

The discovery policy defined in Equation 1 is intentionally general, encompassing both classical pipelines (Figure 1a) and agentic systems (Figure 1b). We provide examples here, and defer specific experiments to Section 4.2.

**Modular Pipelines**   Many discovery strategies follow modular pipelines composed of four interchangeable components:

- **Planner**: Selects compositions to explore, e.g., random, heuristic-, uncertainty- or LLM-based.

- **Generator**: Proposes candidate structures using methods such as AIRSS (Pickard & Needs, 2011), or generative models (Xie et al., 2022; Park et al., 2025; Zeni et al., 2025; Antunes et al., 2024)

- **Filter**: Drops low-quality candidates from the generator e.g. those that are chemically invalid or redundant, e.g., SMACT (Davies et al., 2016)), structural uniqueness via `pymatgen.StructureMatcher` (Ong et al., 2013), or simple geometric constraints such as minimum interatomic distances.

- **Selector**: Ranks and selects from generated candidates for selection using e.g., heuristics, surrogate models such as MLIPs (Batatia et al., 2023; Rhodes et al., 2025), or with LLMs.

**Agentic Systems**   MADE supports fully agentic systems in which an LLM autonomously orchestrates the discovery loop via tool use, internal state tracking, and multi-step reasoning to choose the next structure to evaluate (Jia et al., 2024; Badrinarayanan et al., 2025; Inizan et al., 2025).

## 4. Experiments

We demonstrate the MADE framework by using it to benchmark end-to-end materials discovery on a variety of policies, comparing the contributions of individual components in deterministic pipelines, and end-to-end agentic systems. We then study how discovery performance scales with chemical system complexity and stability thresholds. Full implementation details and extended results are provided in Appendices B and C.

### 4.1. Benchmark Environments

We evaluate discovery performance across multiple chemical systems and random seeds. For each system, we run 5 independent discovery episodes with an oracle query budget of 50. Unless otherwise stated, results are averaged over 10 systems for each of ternary, quaternary, and quinary intermetallic chemical spaces (full list in Appendix B.3). These spaces are especially relevant given recent interest in high-entropy alloys as a relatively unexplored but potentially fruitful search space for various applications (Nakaya & Furukawa, 2024; Yang et al., 2025).

While we focus on intermetallics as a case study in the main results, we investigate generalization across additional commercially relevant chemical spaces (chalcogenides, halides and oxides) and note similar trends in performance across policies in Appendix C.6.

*Table 1.* Results for discovery policies averaged across all system sizes and episodes, at a 0.1 eV stability threshold with a query budget of 50. **Higher** is better for all columns. The error in the final significant figure is given in brackets as the standard error in the mean. Statistically significant top results are highlighted in bold. Details on metrics and experimental setup are given in Sections 3.3 and 4.

| Policy | | | Discovery Performance | | | | Discovery Diversity | | |
|---|---|---|---|---|---|---|---|---|---|
| Generator | Planner | Selector | AF | EF | AUDC | mSUN | Mean Comp. L1. | Unique Comps. | Unique SGs |
| Random | Random | Random | 1.00(0) | 1.00(0) | 0.12(1) | 0.12(1) | **0.98(4)** | 6.3(5) | 1.00(0) |
| Random | Diversity | Random | 1.0(1) | 1.11(6) | 0.11(1) | 0.12(1) | **0.97(4)** | 6.6(5) | 1.00(0) |
| Random | LLM | Random | 1.2(1) | 1.5(1) | 0.123(9) | 0.124(9) | 0.48(3) | 5.5(3) | 1.00(0) |
| Chemeleon | Random | Random | 1.7(1) | 1.70(9) | 0.19(2) | 0.19(2) | 0.89(3) | 10.4(7) | 2.7(1) |
| Chemeleon | Diversity | Random | 2.1(2) | 1.97(9) | 0.19(2) | 0.21(2) | **0.96(3)** | 10.8(8) | 3.7(2) |
| Chemeleon | LLM | Random | 3.9(4) | 3.3(2) | 0.27(2) | 0.26(2) | 0.70(2) | 10.4(5) | 6.8(3) |
| Chemeleon | – | MLIP | **6.4(5)** | 5.3(4) | **0.42(2)** | **0.39(2)** | 0.84(2) | **19(1)** | 3.3(1) |
| LLM Orch. | – | – | 5.4(5) | **6.0(4)** | **0.40(2)** | **0.40(2)** | 0.71(3) | 10.6(4) | **10.4(3)** |

**Chemical System and Initial Materials** For each chemical system, we construct $H_0$ using structures retrieved from Materials Project with a maximum of 20 atoms in the unit cell (MP-20) via the API (Horton et al., 2025). We recompute formation energies of structures in $H_0$ using the oracle. A stability threshold of 0.1 eV/atom is used by default, with tighter thresholds (0.01 eV/atom) explored in Section 4.4.

**Oracle** We use a state-of-the-art MLIP (`orb-v3-conservative-inf-omat`) (Rhodes et al., 2025) as the formation energy oracle. All structures were relaxed (including unit cell parameters) following the same optimization configuration used in Matbench Discovery (Riebesell et al., 2025).

**Structure Matching** Structural uniqueness is computed using `pymatgen.StructureMatcher` on the primitive cell of each structure with the default lattice, site, and angle tolerances (`ltol=0.2`, `stol=0.3`, `angle tol=5.0`). This is used to compute novelty and uniqueness metrics.

## 4.2. Discovery Policies and Pipeline Components

As in Section 3.5, we categorize strategies by component to isolate their impact on discovery efficiency and diversity. We note that we do not attempt to exhaustively benchmark all possible strategies and generative models. Instead, we show that our framework enables comparison of the utility of each component in an end-to-end pipeline. More details on specific policy implementations are given in Section B.4.

### 4.2.1. PLANNERS

**Random** A composition is selected uniformly at random from the allowed compositional space, without regard to prior evaluations. This provides a non-adaptive baseline.

**Diversity** To encourage exploration, the diversity planner selects the composition that maximizes the minimum Euclidean distance (in normalized composition space) to previously evaluated compositions and $H_0$. This biases sampling toward unexplored regions of composition space.

**LLM-based Planning** The LLM planner uses an LLM (GPT 5.1) to adaptively select compositions. The planner is prompted with previously explored compositions and oracle feedback, and proposes the next composition to explore, balancing exploitation of promising regions against exploration of new compositional space.

### 4.2.2. STRUCTURE GENERATORS

**Random** Atoms are placed uniformly at random in fractional coordinates, with lattice parameters sampled from $U(3, 15)$ Å, and angles from $U(60, 120)°$. This generator provides an uninformed structure proposal baseline.

**Chemeleon** We use Chemeleon (Park et al., 2025) trained on MP-20 as an example of a generative model for crystal structure prediction. Chemeleon produces plausible crystal structures similar to its training data distribution, providing a strong prior for generating stable structures.

**Further Generative Model Ablations** We also benchmarked OMatG (Höllmer et al., 2025) and MatterGen (Zeni et al., 2025), two of the top performing models on the LeMat-GenBench leaderboard (Betala et al., 2025). These additional results are given in Appendix C.4.

### 4.2.3. SELECTORS

**Random** A structure is chosen uniformly at random from the generations.

**MLIP** We use a lower fidelity[2] MLIP, `MACE-MP-0-medium` (Batatia et al., 2023) as a surrogate model for ranking candidates. This enables a similar comparison to see how MLIP rankings speed up discovery. Unlike in the other strategies, we generate a large batch (1024) of structures from across the phase diagram (instead of deciding on a specific composition first) and rank them using the MLIP, mirroring Matbench Discovery.

### 4.2.4. AGENTIC LLM ORCHESTRATOR

Finally, we evaluate a fully agentic LLM-based discovery policy (LLM Orch.) implemented using a ReAct-style control loop (Yao et al., 2022). At each iteration, the agent conditions on the complete history of evaluated structures and their stability outcomes, together with a summary of previously generated but unevaluated candidate structures stored in an internal buffer. The agent iteratively selects actions from a fixed tool set, including composition selection, conditional structure generation (using Chemeleon or direct structure creation), MLIP-based scoring, and flexible buffer querying. This formulation allows the agent to adapt composition-level exploration and structure-level refinement based on accumulated feedback.

After each oracle evaluation, the buffer and evaluation history are updated and used to inform subsequent decisions. Unlike fixed pipelines, which apply a predetermined sequence of steps, the orchestrator dynamically interleaves generation, scoring, and selection based on the current buffer state and prior evaluations, enabling evaluation of long-horizon, feedback-driven decision making. This serves as a baseline illustration of agentic coordination using large-context reasoning; more expressive agents could for example integrate literature retrieval, materials databases, or additional surrogates.

We used GPT 5.1 as the base model for the orchestrator results presented in the main text. However, we also benchmarked additional open- and closed-source models, and ablated components of the agent (access to tools, history length etc.) in Appendix C.5.

### 4.3. Results: Discovery against Materials Project

We report averaged metrics over all episodes and system sizes for each discovery policy using a random generator as a baseline in Figure 3 and Table 1.

**Generative models provide strong priors for efficient discovery** As expected, learned generators such as Chemeleon substantially accelerate discovery relative to random baselines, reflecting a strong inductive bias toward plausible, stable structures.

---

[2]As per performance on Matbench Discovery

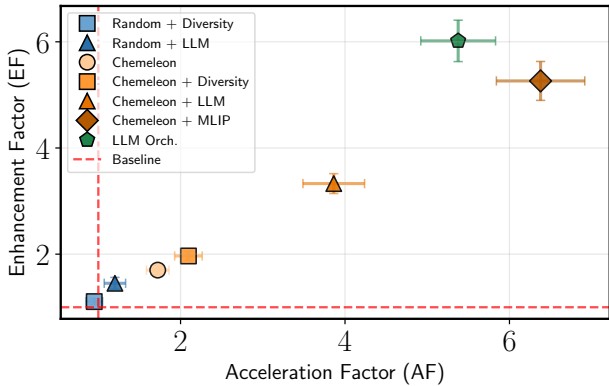

*Figure 3.* End-to-end materials discovery performance of different policies averaged across all system sizes and episodes, against a random generator baseline. Error bars are one standard error in the mean. See Section 4 for details on experimental setup.

**MLIP-based selection significantly accelerates discovery** MLIP-based selection yields the largest single performance gain. The Chemeleon + MLIP pipeline achieves the highest AF among non-agentic methods (AF = 6.4) and the largest AUDC, consistent with prior work demonstrating the effectiveness of surrogate screening in materials discovery.

**Planning accelerates discovery, even with weak generators** Explicit selection of composition spaces to try provides measurable gains beyond generation alone, including in settings with random structure generation. LLM planning in particular achieves significant gains over random acquisition (AF = 1.2, EF = 1.45). When combined with a strong generator, planning yields substantial additional gains: Chemeleon + LLM planning more than doubles performance relative to Chemeleon alone (AF = 3.9).

**End-to-end agentic systems compete with optimized modular pipelines** The fully agentic LLM orchestrator achieves discovery efficiency comparable to the strongest modular pipelines, with significantly improved enhancement factor (EF = 6.0) and competitive AUDC and mSUN (Table 1). While its acceleration factor is slightly lower than the best MLIP-ranked pipeline, the orchestrator consistently discovers a broader range of space groups, indicating a different efficiency–diversity trade-off (Section 4.5). This suggests LLMs can effectively plan and optimize for efficient discovery.

### 4.4. Results: Scaling with System Difficulty

Next we examine how discovery performance changes as the search problem becomes more challenging, focusing on system size (Figure 4) and stability threshold (Figure 5).

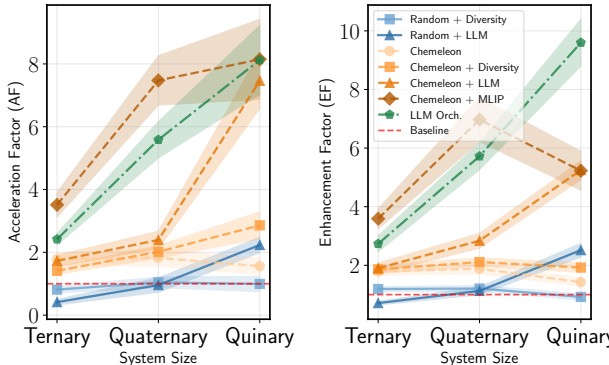

*Figure 4.* Performance of policies at increasing system size. Shaded regions are standard error in the mean across 10 systems with 5 episodes each. We see larger gains for effective planning on larger search spaces over baselines.

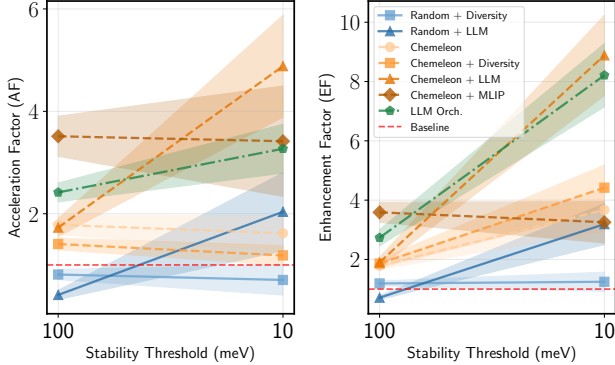

*Figure 5.* Performance of policies at varying stability threshold for discovery. Shaded regions are standard error in the mean across 10 systems with 5 episodes each. Surrogate model rankings (MLIPs) do not generalize well to smaller tolerances due to errors, whereas planning algorithms lead to significant gains over baselines.

**Planning gains importance as system size increases** As the number of constituent elements increases, the number of possible compositions grows combinatorially, making discovery increasingly sparse (Appendix Figure 8). In this regime, adaptive planning yields progressively larger gains over baselines. Figure 4 shows that planning-based strategies, in particular LLM-guided planners and agents, yield progressively larger gains over baselines as system size increases from ternary to quinary systems. Notably, Random + LLM planning outperforms the Chemeleon baseline in larger systems, indicating that composition-level adaptivity can partially compensate for weak generative priors.

**Tighter stability thresholds reduce the effectiveness of surrogate ranking** Figure 5 shows that at stricter stability thresholds, MLIP-based ranking degrades in performance. This is due to surrogate error near the convex hull (Riebesell et al., 2025), highlighting the importance of research in uncertainty-aware MLIP screening for use within acquisition strategies (Coscia et al., 2025; Busk et al., 2023; Betala et al., 2025). In contrast to MLIP rankings, planning-based strategies retain significant gains over baselines, reflecting greater robustness when discovery targets lie close to stability boundaries. This suggests that adaptive exploration becomes increasingly important as the discovery task becomes more selective. In particular, our evidence suggests that LLM-based planners can be less brittle in this regime, likely due to their ability to incorporate broader contextual signals beyond approximate scores from surrogates.

Together, these results highlight that as discovery problems become more challenging, adaptive strategies play an increasingly important role, underscoring the need for benchmarks that evaluate closed-loop discovery behavior.

### 4.5. Results: Diversity of Discovered Materials

Beyond discovery metrics, we also evaluate diversity in the discovered materials using composition- and structure-level metrics (Table 1). In particular, we report the mean pairwise L1 distance between compositions, the number of unique compositions, and the number of unique space-groups (SGs) amongst the discovered mSUN structures.

We find diversity-based planning yields the broadest coverage of composition space, reflected in higher composition distances and expanded phase-diagram coverage (Figures 6 and 7). In contrast, the LLM orchestrator discovers the widest range of SGs, indicating greater structural diversity within explored compositions. These results highlight trade-offs between efficiency and diversity of different strategies.

## 5. Discussion and Conclusions

We introduce MADE, a family of benchmark environments that reframe materials discovery as a closed-loop sequential decision making task, enabling flexible and scalable evaluation of end-to-end discovery beyond what is possible with existing static benchmarks. Using MADE, we show that while pipelines reliant on surrogate model screening perform well for simple systems, adaptive planning strategies become increasingly important as search spaces grow and as surrogate errors become more consequential or out-of-distribution.

**Limitations and Future Work** Current experiments rely on generators and MLIPs trained on Materials Project data, and thus inherit shared distributional biases that may simplify discovery relative to real-world settings. Extending MADE to incorporate DFT or experimental oracles, batched query evaluation, and multi-objective tasks are natural next steps. As a gym-like environment, MADE also enables rein-

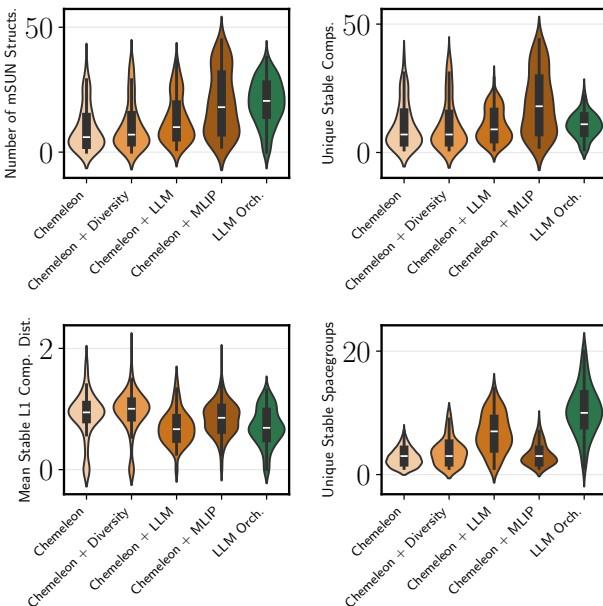

*Figure 6.* Diversity metric distributions for discovered stable structures averaged across system sizes.

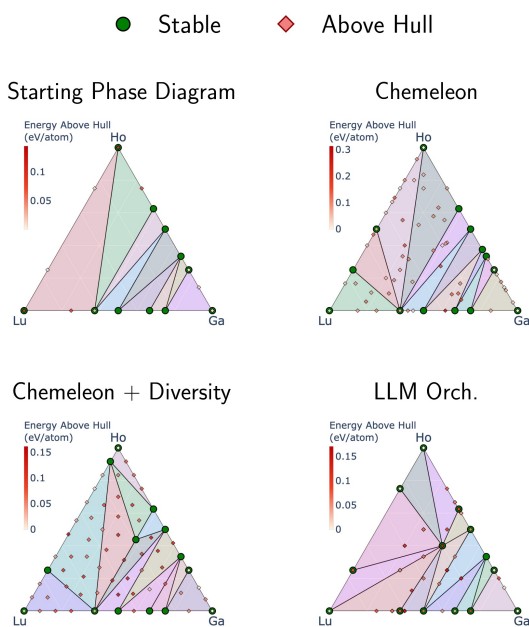

*Figure 7.* Starting (MP-20) and post-acquisition phase diagrams under different strategies on an example ternary system.

forcement learning over the full discovery loop, connecting to recent work on fine-tuning crystal generators and adaptive policies (Chen et al., 2025; Park & Walsh, 2025).

**Outlook**  More broadly, MADE provides a concrete testbed for evaluating core capabilities of agentic systems in realistic scientific discovery settings, including long-horizon planning, reasoning under uncertainty, and learning from feedback. By enabling controlled evaluation of these behaviors in closed-loop environments, MADE can help ground progress toward autonomous scientific discovery systems.

before deployment. More broadly, defining effective evaluation metrics for autonomous discovery agents is critical to ensuring agents pursue human aligned scientific goals.

## Acknowledgments

The authors would like to thank Atılım Güneş Baydin for useful feedback on the paper. SM acknowledges funding from the EPSRC Centre for Doctoral Training in Autonomous Intelligent Machines and Systems (Grant No: EP/S024050/1). YG acknowledges funding from the Turing Fellowship (Grant No. EP/V030302/1). We acknowledge compute support from Modal. This work was funded under the Horizon Europe grant 101213369 DVPS.

## Impact Statement

As aspects of scientific research become increasingly automated, there is a growing need to critically assess the benefits and risks of handing over research autonomy to AI systems. Benchmark frameworks such as the proposed can help surface and study the risks by making agent behavior and decision making processes evident on test-beds

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

# A. Comparison of Materials Discovery Benchmarks

Table 2 provides a high-level comparison of materials discovery benchmarks under the criteria outlined in Section 3.1.

MLIP screening benchmarks (e.g. Riebesell et al. (2025)) that rank candidates in a pool are method-agnostic but are not open-ended; they require a fixed candidate pool set, nor do they incorporate closed-loop feedback.

Generative model benchmarks (e.g. Betala et al. (2025)) enable open-ended generative evaluation but do not incorporate closed-loop feedback or discovery acceleration of end-to-end pipelines.

Prior materials optimization benchmarks (e.g. Abhyankar et al. (2026)) do incorporate feedback but do not provide standardized environments for comparing methods, and often only evaluate against static generative model baselines rather than the model in the context of a discovery pipeline, and only consider top performance of an objective rather than discovery acceleration metrics.

MADE uniquely evaluates discovery acceleration metrics for end-to-end, iterative, and open-ended tasks with oracle feedback, while remaining agnostic to the choice of search or modeling strategy.

*Table 2.* Comparison of MADE with existing classes of materials discovery benchmarks. Legend: ✓ supported, ∼ partially supported, ✗ not supported

| Benchmark Class | End-to-End | Closed-Loop | Open-Ended | Discovery Metrics | Method-Agnostic |
|---|:---:|:---:|:---:|:---:|:---:|
| MLIP screening benchmarks | ✗ | ✗ | ✗ | ∼ | ✓ |
| Generative model benchmarks | ✗ | ✗ | ✓ | ✗ | ✓ |
| Materials optimization benchmarks | ∼ | ✓ | ✓ | ∼ | ∼ |
| **MADE (ours)** | ✓ | ✓ | ✓ | ✓ | ✓ |

# B. Implementation Details

## B.1. General Computational Details

**Configuration and Reproducibility**    Pipelines, components, and environments are all instantiated through declarative YAML configuration files using Hydra (Yadan, 2019), enabling controlled comparison of strategies by changing a single module while holding others fixed. This design promotes reproducibility by making all experimental choices (models, budgets, seeds, thresholds etc.) explicit and versionable, and lowers the barrier to implementing and evaluating new components within the same experimental protocol. Code, configurations, and scripts used to generate the results in this paper are publicly released in the code repository[3].

**Compute**    Experiments were run on a single NVIDIA T4 GPU per episode. Episodes were parallelized across systems and random seeds using Modal cloud compute.

**LLM Usage**    We use DSPy (Khattab et al., 2023) as the backend for structured prompt design and outputs. In DSPy, prompts are defined in through inputs and output typed signatures rather than free-form text prompts. Each signature specifies named input fields (e.g., evaluation history) and output fields, which DSPy automatically serializes into a consistent prompt format for the language model. This approach enables reproducible prompt construction, modular composition of reasoning components (such as chain-of-thought (Wei et al., 2022) or ReAct (Yao et al., 2022)), and systematic control over what information is exposed to the model at each step.

## B.2. MADE Pseudocode

Listing 1 contains pseudocode for the base classes of the MADE benchmark. The Oracle, Environment and Agent base classes are flexible to allow for different methods to be implemented. We make use of `pymatgen` (Ong et al., 2013) classes to use phase diagrams as the environment state and convex hull computations.

---

[3]https://github.com/diffractivelabs/MADE

*Listing 1.* Pseudocode for key classes in MADE

```
class Oracle:
    def __init__(self, model):
        self.model = model  # e.g., MLIP, DFT, experimental oracle

    def predict_energy(self, structure):
        energy = self.model.predict(structure)
        return energy

class Environment:
    def __init__(self, oracle, initial_known_structures, chemical_system):
        self.oracle = oracle
        self.chemical_system = chemical_system
        self.known_structures = initial_known_structures
        self.energies = {s: self.oracle.predict_energy(s)
                            for s in initial_known_structures}
        self.update_convex_hull()

    def step(self, structure):
        energy = self.oracle.predict_energy(structure)
        self.known_structures.append(structure)
        self.energies[structure] = energy
        self.update_convex_hull()
        return energy

    def update_convex_hull(self):
        self.convex_hull = ConvexHull(self.known_structures, self.energies)

    def reset(self):
        self.known_structures.clear()
        self.energies.clear()
        self.update_convex_hull()

class Agent:
    def __init__(self, chemical_system):
        self.chemical_system = chemical_system

        self.policy = Policy(chemical_system)

    def predict_next_structure(self, env):
        # policy can use existing convex hull and query history to propose next structure
        next_structure = self.policy(env)
        return next_structure

# example rollout
oracle = Oracle(model)
env = Environment(oracle, initial_known_structures, chemical_system)
agent = Agent(chemical_system)

for t in range(query_budget):
    structure = agent.predict_next_structure(env)
    energy = env.step(structure)
```

## B.3. Benchmark Environments

**Chemical Systems**     As described in Section 4.1, we primarily evaluate discovery across intermetallic chemical systems of increasing complexity. We randomly sample 10 systems from each of ternary, quaternary, and quinary element spaces (3–5 elements), excluding systems with radioactive elements. The systems used are shown in Table 3, where we additionally report the chalcogenide, halide and oxide systems used for additional evaluation in Appendix C.6. Each system is evaluated

*Table 3.* Chemical systems used for benchmarking, grouped by chemistry class. intermetallic systems are used for results in the main text; chalcogenide, halide, and oxide systems are used for further results reported in Appendix C.6.

| Intermetallics | | | Chalcogenides | Halides | Oxides |
|---|---|---|---|---|---|
| **Ternary** | **Quaternary** | **Quinary** | | | |
| Mg–Sn–Sr | Ba–Nd–Ni–W | Cd–Li–Nd–Ti–W | Cu–In–Se | Li–In–Cl | Bi–Fe–O |
| Au–K–Tb | Eu–Nb–Sn–Tl | Cd–Gd–Mn–Na–Ta | Zn–In–S | Li–Er–Cl | Cu–Fe–O |
| Co–Mg–Na | Ce–Er–Pb–Rh | Co–Hg–Mg–Sr–W | Ge–Sb–Te | Li–Lu–Cl | Li–Co–O |
| Hf–Ni–Zr | Dy–K–Pd–Sm | Ca–Fe–Gd–Pb–Tb | Bi–Sb–Te | Li–Y–Cl | Li–Mn–O |
| Co–Dy–W | Co–Dy–Ta–Y | Al–Hg–K–Mg–W | Cu–Zn–Sn–S | Cs–Ag–Bi–Br | Li–La–Zr–O |
| Ga–Pt–Tm | Au–Cr–Cs–Dy | Ag–Nd–Pd–Pt–Tb | Cu–Zn–Sn–Se | Cs–Na–Bi–Cl | Ba–Sr–Ti–O |
| Ga–Ho–Lu | Ca–Pd–Sn–W | Al–Lu–Pt–Rb–Sm | Li–Ge–P–S | Cs–Ag–Bi–Cl | Bi–Fe–Ti–O |
| Al–Li–V | Au–Tb–V–Y | Co–Hf–In–Ru–Tm | Cu–In–Ga–Se–S | Rb–Cs–Ag–Bi–Br | Li–Ni–Mn–Co–O |
| Al–V–Zn | Ce–Ir–Pt–Sn | Ho–In–Mg–Pd–Zr | Ag–Bi–Sb–Te–Se | Cs–Na–Sc–Br–Cl | Ba–Zr–Ce–Y–O |
| Co–Pd–Tl | Ba–Be–Hf–Li | Cr–Fe–Lu–Pt–Sc | Mg–Sb–Bi–Te–Se | Cs–Na–Ce–Br–Cl | Na–Fe–Mn–Ni–O |

over 5 independent discovery episodes.

**Oracle**  As mentioned in Section 4.1, We use `orb-v3-conservative-inf-omat` (Rhodes et al., 2025) as the formation energy oracle. All structures were relaxed (including unit cell parameters) using the FIRE optimizer (Bitzek et al., 2006) for a maximum of 500 steps or fmax of 0.02 in ASE (Larsen et al., 2017) before evaluating the final energy, mirroring Matbench Discovery (Riebesell et al., 2025).

**Note on Discovery Metrics**  If, for a given system, the baseline policy does not find any new structures, the acceleration factor is not defined. In these settings, we define the acceleration factor to be equal to the maximum number of queries (50). This was the case for a small number of quinary systems.

### B.4. Specific Policy Details

**General Workflow for Non-Agentic Policies**  Unless otherwise stated, each oracle query proceeds by first selecting a single composition using the planner, then generating a batch of 32 candidate structures conditioned on that composition using the generator, from which a single structure is selected for evaluation using the selector.

#### B.4.1. RANDOM GENERATOR

The random generator generates crystal structures by randomly assigning lattice parameters $a, b, c$ from the uniform distribution $U(3, 15)$ Å, and angles $\alpha, \beta, \gamma$ from $U(60, 120)$ degrees. and fractional atomic positions from $U(0, 1)$. While more sophisticated heuristics could adapt lattice constants to unit-cell size, we use this simple formulation to provide a minimal baseline.

#### B.4.2. CHEMELEON GENERATOR

We use Chemeleon (Park et al., 2025) trained on MP-20 to produce structure proposals conditioned on composition. For planner-based policies, we generate 32 candidate structures for the selected composition at each query step. For the MLIP-ranked policy, we generate 1024 structures across randomly sampled valid compositions to mimic common high-throughput generative screening workflows. This roughly matches the same total number of generations over the episode as the sequential planning setting ($50 \times 32$) for fair comparison.

#### B.4.3. CHEMELEON + MLIP RANKING

For MLIP-based baselines, we generate a large batch (1024) of candidate structures across the phase diagram and rank all candidates using a lower-fidelity MLIP (MACE `MP-0-medium` (Batatia et al., 2023)) surrogate before oracle evaluation. This mirrors common MatBench-style discovery pipelines and isolates the effect of surrogate-based ranking without adaptive planning.

### B.4.4. DIVERSITY PLANNER

The diversity planner selects compositions to maximize coverage of composition space while accounting for prior exploration outcomes. All compositions up to a maximum stoichiometry are enumerated at initialization. Each composition is represented as a vector of fractional elemental concentrations, and pairwise distances (Euclidean by default) are computed between candidate compositions and a reference set consisting of previously attempted compositions and elemental end members.

Each composition $c$ is represented by a normalized composition vector

$$\mathbf{x}_c = \big(x_{c,1}, x_{c,2}, \ldots, x_{c,d}\big),$$

where $x_{c,i}$ denotes the fractional concentration of element $i$ and $d$ is the number of elements in the chemical system. Let $\mathcal{R}$ denote the reference set of compositions, consisting of all previously attempted compositions together with elemental end members. We then define the valid reference set for a composition as,

$$\mathcal{R}_c = \{c' \in \mathcal{R} \mid \text{red}(c') \neq \text{red}(c)\},$$

where $\text{red}(\cdot)$ denotes the reduced chemical formula. This mask helps avoid the minimum distances being trivially zero for the same reduced compositions. The diversity distance for composition $c$ is then defined as the minimum Euclidean distance to this masked reference set:

$$D(c) = \min_{c' \in \mathcal{R}_c} \|\mathbf{x}_c - \mathbf{x}_{c'}\|_2 = \min_{c' \in \mathcal{R}_c} \sqrt{\sum_{i=1}^{d} (x_{c,i} - x_{c',i})^2}. \tag{4}$$

The diversity score is multiplied by a composition-specific weight that encodes exploration history. Unattempted compositions receive a fixed weight (5.0), strongly prioritizing unexplored regions of composition space. For previously attempted compositions, the weight is computed as

$$w(c) = \alpha \cdot \frac{1}{n_c + 1} + \beta \cdot (1 - r_c), \tag{5}$$

where $n_c$ is the number of attempts for composition $c$, $r_c$ is its empirical success rate, and $(\alpha, \beta) = (0.7, 0.3)$. The planner selects the composition with the highest weighted diversity scores $w(c)D(c)$, encouraging systematic exploration of sparse regions while still revisiting compositions with unresolved failures if all compositions have been attempted.

### B.4.5. LLM PLANNER

The LLM planner operates at the composition level, selecting which compositions to explore based on accumulated oracle feedback. At each planning step, the raw environment state is summarized into a structured context dictionary, which is then passed to a DSPy signature and automatically serialized into a prompt.

Concretely, the planner's input fields include: (i) the allowed chemical elements defining the search space, (ii) the stability threshold and query count, (iii) a summarized list of previously evaluated entries sorted by energy above the convex hull (including reduced and full formulas, energies, and stability labels), (iv) composition-level trial counts for both reduced formulas (phase diagram points) and full formulas (unit cell sizes), and (v) the most recent query and observation. To control context length, the number of summarized entries can be capped, with stable or metastable entries always included and the remainder randomly sampled. We place the cap at 20 structures. The planner produces a single structured output: a list of candidate compositions specified as full formulas with explicit unit cell sizes, which are subsequently validated against element and stoichiometry constraints before structure generation. The system prompt used for the LLM planner is provided in Listing 2.

*Listing 2.* LLM Planner Prompt

```
You are a planner for a material discovery experiment. Your goal is to discover as
    many NOVEL, UNIQUE, STABLE (or metastable) structures as possible.

CRITICAL CONSTRAINT: You MUST ONLY use elements from the provided 'elements' list in
    your compositions. And you MUST ONLY propose compositions that are within the
    max_stoichiometry.
```

```
        – Example: If elements=['Li', 'O'], you can propose Li2O, LiO2, etc., but NOT Na2O,
            Fe2O3, etc.
        – Example: If max_stoichiometry=20, you can propose Li2O, LiO2, etc., but NOT Li19O19.

        DEFINITIONS:
        – STABLE/METASTABLE: Structures with e_above_hull <= stability_tolerance.
        – NOVEL: Not already known on the convex hull (is_newly_discovered=True).
        – UNIQUE: Structurally distinct from previously evaluated structures.
        – Entries marked is_stable_or_metastable=True with is_newly_discovered=True are
            successful discoveries.

        PHASE DIAGRAM CONCEPTS:
        – Reduced formulas (e.g., Li2O) represent UNIQUE POINTS on the phase diagram.
        – Different reduced compositions = different phase diagram points, then PRIORITIZE
            DIVERSE reduced compositions for broad coverage.

        STRUCTURE DIVERSITY AT SAME COMPOSITION:
        – Multiple DIFFERENT structures can exist at the SAME reduced composition (same phase
            diagram point)
        – Different structures for the SAME composition can have DIFFERENT stabilities (one
            unstable doesn't mean all are!)
        – Different unit cell sizes (Li2O vs Li4O2) create different structures but occupy the
            SAME phase diagram point

        STRATEGY:
        – Explore diverse reduced compositions (different phase diagram points) to maximize
            phase diagram coverage.
        – If a reduced composition has yielded stable/metastable NOVEL structures, consider
            proposing MORE unit cell sizes for it as additional stable polymorphs may exist.
        – Compositions with only [unstable] entries may still have stable structures.
        – Balance exploration (new reduced compositions) vs exploitation (trying to find
            stable structures for compositions with only [unstable] entries)

        Propose FULL formulas with specific unit cell sizes (e.g., Li2O, Li4O2, LiO2) not just
            reduced formulas.
```

### B.4.6. AGENT LLM ORCHESTRATOR

We implement the agentic discovery policy using DSPy's `ReAct` framework (Khattab et al., 2023; Yao et al., 2022). At each oracle query, the orchestrator reasons over the current discovery state and selects actions from a fixed tool set. The agent maintains a composition-indexed buffer of candidate structures that have passed static validity checks; a uniqueness filter is always re-applied to new generations, and structures are cached by hash to avoid duplicate processing.

The LLM has access to (i) a summary of the current buffer, (ii) a bounded evaluation history (most recent 20 oracle queries), and (iii) known stable materials from the phase diagram. Each decision step is limited to 10 ReAct iterations.

**Available tools.** The orchestrator can invoke the following tools:

- `generate_structures`: generate candidate structures for specified compositions using the generators defined above (Random or Chemeleon);

- `create_structure`: directly specify a crystal structure by explicitly defining lattice parameters and atomic positions;

- `score_buffer`: score buffered structures using a selected scorer (e.g., diversity, MLIP, or LLM-based, as defined above.);

- `list_compositions`: list compositions in buffer ordered by count or score;

- `query_structures`: retrieve all, random, top or bottom $k$ scoring structures for a given composition;

- `get_buffer_stats`: report buffer statistics for situational awareness;

- `select_for_evaluation`: select a single structure (by specifying a composition and buffer index) for oracle evaluation. This is always the final tool call.

The orchestration system prompt is given in Listing 3.

*Listing 3.* LLM Orchestrator Prompt

```
You are an autonomous materials discovery agent.

OBJECTIVE: Find as many NOVEL, UNIQUE, STABLE (or metastable) structures as possible.
Use the available tools, then select ONE composition + structure for oracle evaluation.

- Structures with e_above_hull <= stability_tolerance are stable/metastable (SUCCESS!)
- Entries marked [STABLE, NOVEL] in evaluation_history are successful discoveries
- We want to MAXIMIZE the number of novel stable structures found

IMPORTANT: Different structures for the SAME composition can have DIFFERENT stabilities.
- One unstable structure for a composition does NOT mean all structures for that
    composition are unstable.
- Generators produce many different structures for the same composition

UNIT CELL SIZE MATTERS:
- Compositions are stored by REDUCED formula (e.g., Li2O)
- But you can generate different UNIT CELL SIZES: Li2O, Li4O2, Li6O3, etc.
- These occupy the same position on the phase diagram but are different structures

BUFFER ORGANIZATION:
- Buffer is organized by reduced formula: {composition: [structures]}
- Each structure shows its full formula (unit cell size) and index
- Selection is two-step: pick composition, then pick structure index

WORKFLOW:
1. Decide which composition(s) to explore based on evaluation history
2. Generate or create candidate structures for those compositions
3. Score candidates if needed to prioritize within each composition
4. List compositions and query structures to decide what to evaluate
5. Select ONE composition + structure for oracle evaluation

STRATEGY GUIDANCE:
- If buffer empty/small: generate more structures
- If buffer has candidates: score, query, and select
- Balance exploration (new compositions) vs exploitation (promising ones)
```

### B.4.7. FILTERS

Generated structures can be passed through inexpensive validity filters prior to oracle evaluation to mirror real discovery pipelines. Specifically, we implement:

- a minimum interatomic distance filter to remove unphysical atomic overlaps. This is set at 0.5 Å between atoms, as per the structural validity evaluation metric used in MatterGen (Zeni et al., 2025).

- a uniqueness filter that filters previously attempted structures (see 3.4).

- a chemical validity filter based on SMACT constraints. For intermetallic systems, this is redundant.

These filters were applied to the LLM orchestrator and MLIP ranking policies. We observed no qualitative changes in relative performance when applying these filters to other baselines versus not using them, so we do not explicitly compare results with and without the filters.

# C. Further Results

## C.1. Scaling System Complexity

Figure 8 illustrates how the number of valid compositions grows rapidly with system size. For each composition, there exists a vast space of possible crystal structures arising from different lattice symmetries, atomic arrangements, and unit-cell sizes. This combinatorial growth in both composition and structure leads to an extremely sparse discovery landscape at larger system sizes, making naive enumeration infeasible and underscoring the need for effective, adaptive search strategies to identify new stable materials.

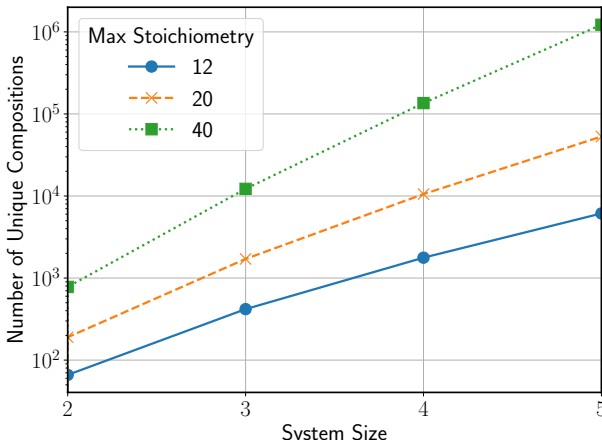

*Figure 8.* The number of unique compositions that can be explored grows rapidly with system size.

## C.2. Distributional Results and Additional Structural Diversity Metrics

Figure 9 shows distributions of the summary results given in Table 1. The top row gives the discovery metrics while the bottom row gives diversity metrics.

We additionally report a continuous structural discrepancy metric: the average minimum distance (AMD) (Widdowson et al., 2022; Widdowson & Kurlin, 2022). We report the mean pairwise AMD between discovered mSUN structures. Naturally, the random generator produces structures with large discrepancies.

## C.3. Results Across System Sizes

Figures 10, 11 and 12 show discovery metrics broken down by system size and policy. We report distributions over all metrics per system size in Figures 13, 14 and 15.

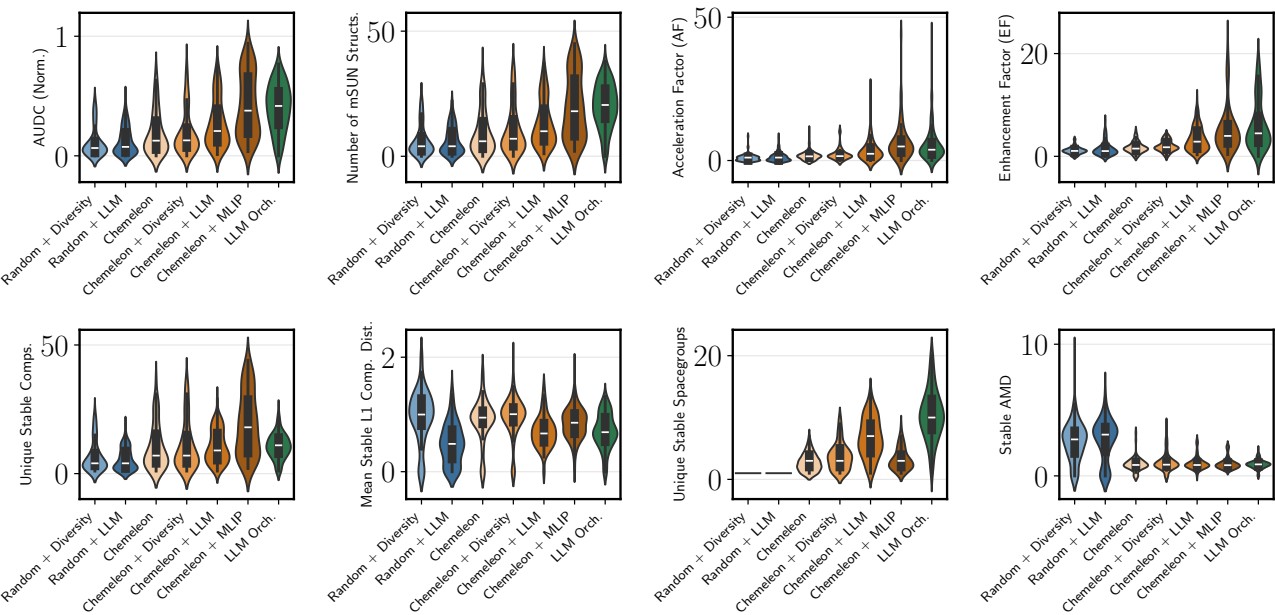

*Figure 9.* Discovery and diversity metric distributions, aggregated over system sizes.

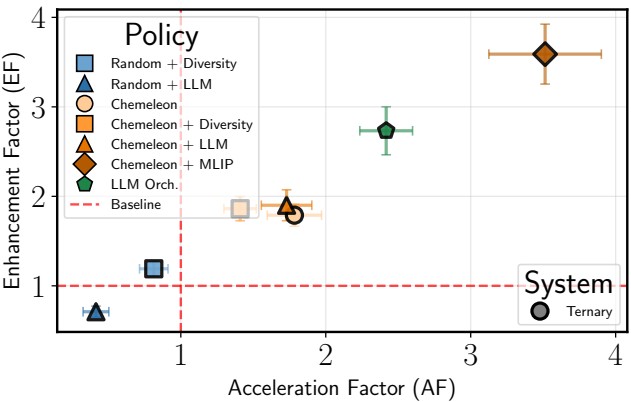

*Figure 10.* End-to-end materials discovery performance of different policies on ternary systems. The error bars are standard errors in the mean over 10 systems, each with 5 episodes.

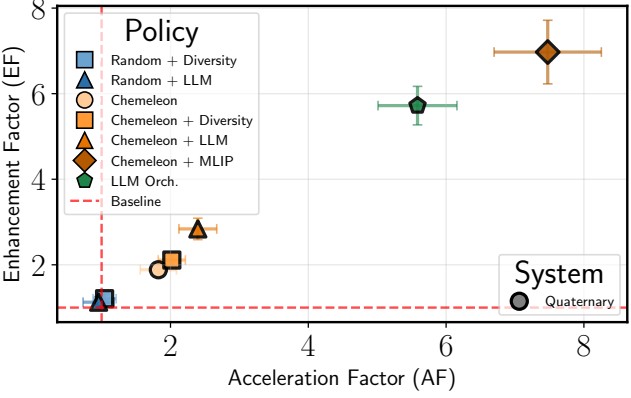

*Figure 11.* End-to-end materials discovery performance of different policies on quaternary systems. The error bars are standard errors in the mean over 10 systems, each with 5 episodes.

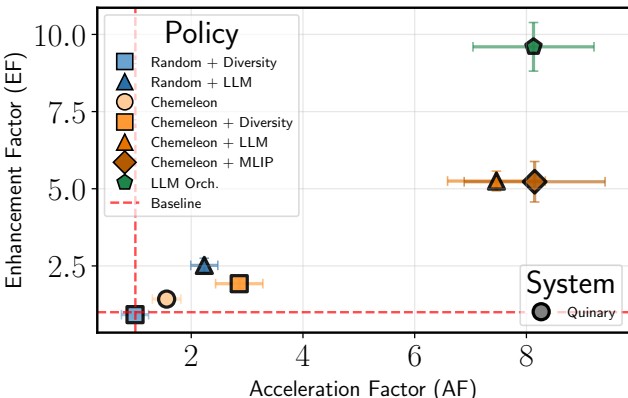

*Figure 12.* End-to-end materials discovery performance of different policies on quinary systems. The error bars are standard errors in the mean over 10 systems, each with 5 episodes.

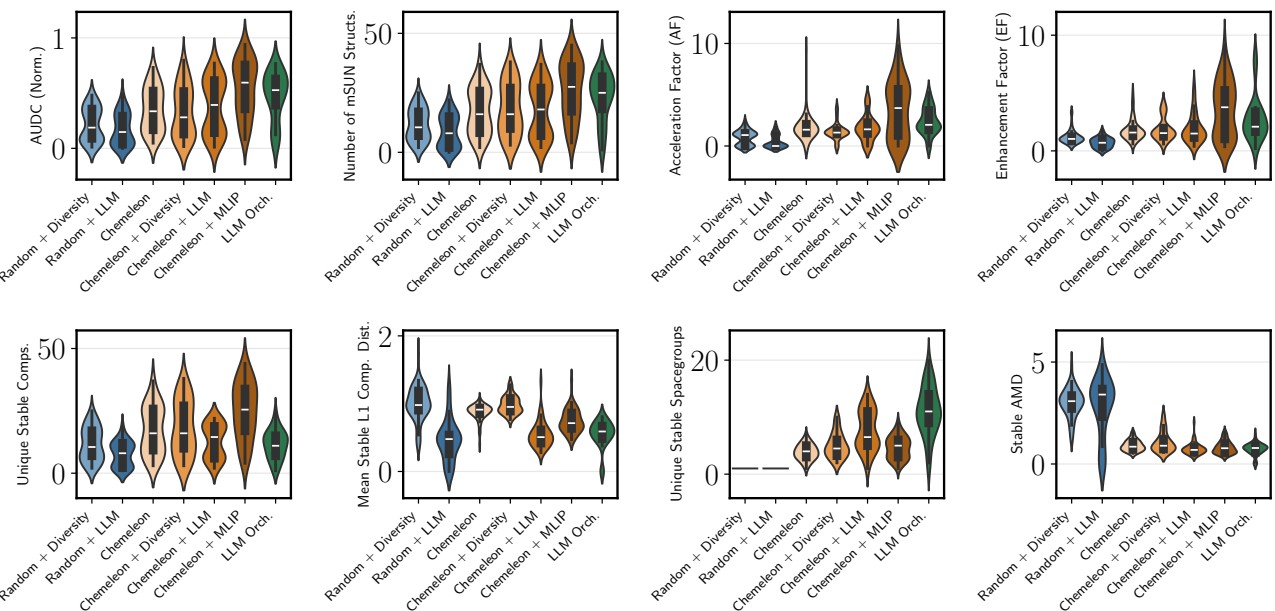

*Figure 13.* Discovery and diversity metric distributions for experiments on ternary systems.

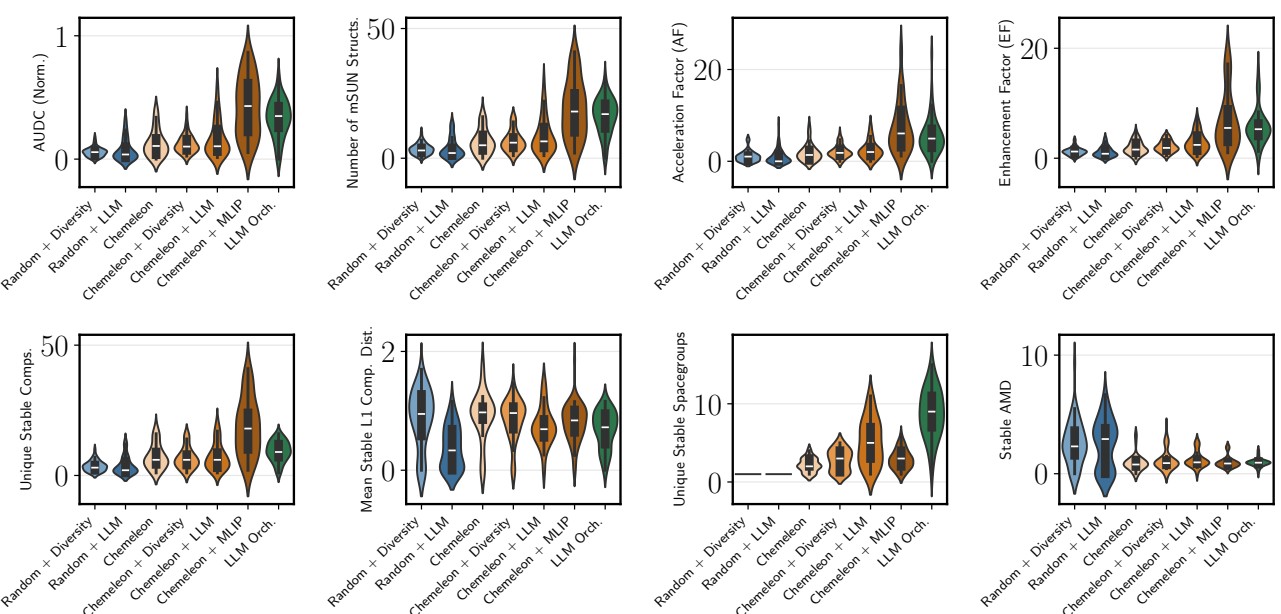

*Figure 14.* Discovery and diversity metric distributions for experiments on quaternary systems.

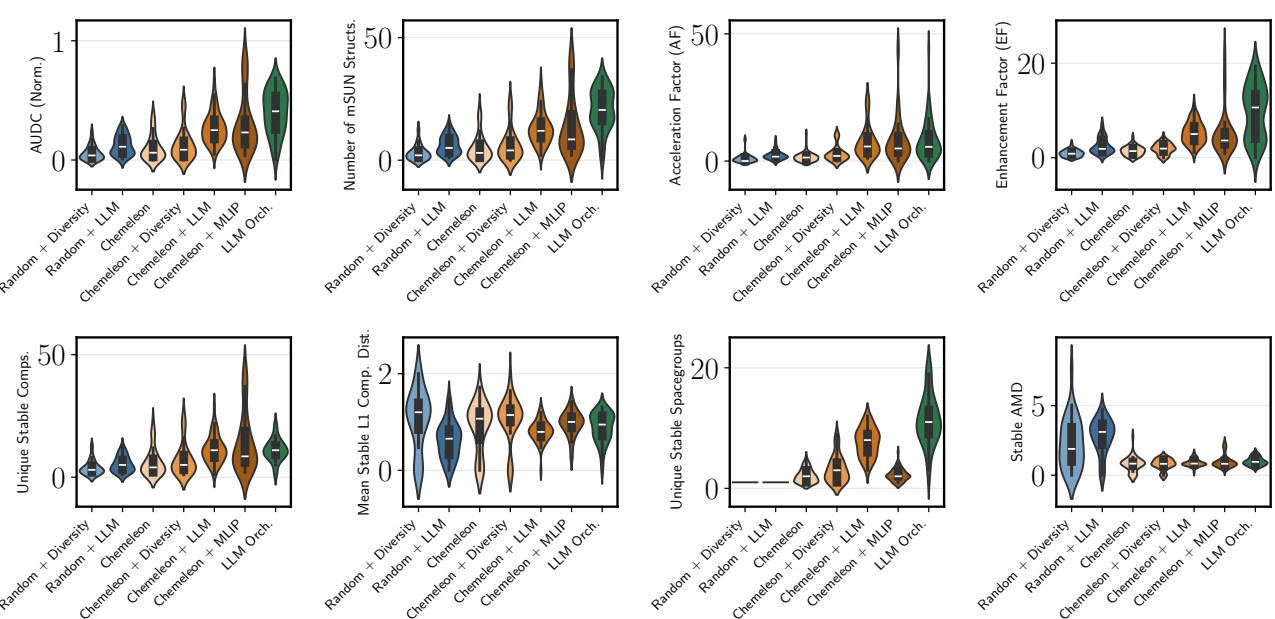

*Figure 15.* Discovery and diversity metric distributions for experiments on quinary systems.

## C.4. Ablating Generative Models

In addition to Chemeleon, we benchmarked further state of the art generative models (MatterGen (Zeni et al., 2025) and OMatG (Höllmer et al., 2025)) to highlight the flexibility of the framework. The same experimental set up as in Section 4.1 was used. The results are shown in Table 4. We note that these model checkpoints have been trained on more data than $H_0$, so there is risk of recall of additional known stable structures.

**MatterGen (Zeni et al., 2025)**   We use the official implementation[4] and the `chemical_system` checkpoint[5] to condition generations to be within the chemical system being evaluated. As it is not possible to directly condition on specific compositions, we only evaluate the random selector policy. We use a diffusion guidance factor of 2.

**OMatG (Höllmer et al., 2025)**   We use the official implementation[6] and the `Alex-MP-20-CSP` checkpoint[7], using the recommended configs.

*Table 4.* Generative model ablation results averaged across all system sizes and episodes, at a 0.1 eV stability threshold with a query budget of 50. **Higher** is better for all columns. The error in the final significant figure is given in brackets as the standard error in the mean. Details on metrics and experimental setup are given in Sections 3.3 and 4.

| Policy | | Discovery Performance | | | | Discovery Diversity | | |
| --- | --- | --- | --- | --- | --- | --- | --- | --- |
| Generator | Planner | AF | EF | AUDC | mSUN | Mean Comp. L1. | Unique Comps. | Unique SGs |
| Chemeleon | Random | 1.7(1) | 1.70(9) | 0.19(1) | 0.19(1) | 0.89(3) | 10.4(7) | 2.7(1) |
| Chemeleon | Diversity | 2.1(2) | 2.0(1) | 0.19(2) | 0.21(2) | **0.96(3)** | 10.8(8) | 3.7(2) |
| Chemeleon | LLM | **3.9(4)** | **3.3(2)** | **0.27(2)** | **0.26(2)** | 0.70(2) | 10.4(5) | 6.8(3) |
| OMatG | Random | 1.8(2) | 1.71(8) | 0.19(1) | 0.18(1) | 0.90(3) | 9.6(7) | 1.05(2) |
| OMatG | Diversity | 1.9(2) | 1.88(9) | 0.18(1) | 0.19(1) | **0.98(3)** | 10.0(7) | 1.01(1) |
| OMatG | LLM | 3.0(4) | 2.6(2) | 0.21(1) | 0.21(1) | 0.66(3) | 9.9(5) | 1.13(3) |
| MatterGen | Random | 2.7(3) | 2.4(2) | 0.26(2) | 0.24(2) | 0.43(2) | **11.4(6)** | **8.1(4)** |

## C.5. Ablating LLM Models and Agent Components

In Table 5 we show results for an ablation study on agent components and LLM model providers for the LLM orchestrator policy. The same experimental set up as in Section 4.1 was used.

**Tooling Ablation**   Removing the MLIP scorer actually increases performance, meaning that agentic planning can be more effective than surrogate ranking that could lead the agent astray. Interestingly, only small reductions in performance are observed in the absence of generative tools. LLM proposed structures are generally reasonable.

**Reasoning Steps**   Performance scales with reasoning depth (number of ReAct steps), and not as significantly with memory (number of structures in the history context).

**Structural Context**   Performance degrades without structural information as expected, as the energy of a system is highly structure dependent.

**LLM Providers**   Gains vary across model sizes/costs (GPT 5.4 Nano) and providers (Anthropic, DeepSeek), but all show speed ups compared to the baselines (AF, EF significantly greater than 1).

---

[4] https://github.com/microsoft/mattergen
[5] https://huggingface.co/microsoft/mattergen
[6] https://github.com/FERMat-ML/OMatG
[7] https://huggingface.co/OMatG/Alex-MP-20-CSP

*Table 5.* Ablation results for the LLM orchestration policy averaged across intermetallic systems and episodes, at a 0.1 eV stability threshold with a query budget of 50. **Higher** is better for all columns. The error in the final significant figure is given in brackets as the standard error in the mean. The default configuration uses GPT 5.1 with ReAct $k = 10$, history $k = 20$, structural information, MLIP and Chemeleon tools. Details on metrics and experimental setup are given in Sections 3.3 and 4.

| LLM Orchestrator | Discovery Performance | | | | Discovery Diversity | | |
|---|---|---|---|---|---|---|---|
| | AF | EF | AUDC | mSUN | Mean Comp. L1. | Unique Comps. | Unique SGs |
| *Agent Component Ablation* | | | | | | | |
| Default | 5.4(5) | 6.0(4) | 0.40(2) | 0.40(1) | 0.71(3) | 10.6(4) | 10.4(3) |
| 5 ReAct Steps | 5.4(4) | 6.1(4) | 0.41(2) | 0.40(1) | 0.71(2) | 11.0(4) | 10.3(3) |
| 20 ReAct Steps | 5.9(6) | 6.0(4) | 0.40(2) | 0.40(1) | 0.68(3) | 10.8(4) | 10.0(3) |
| 5 History Steps | 5.2(5) | 6.0(4) | 0.40(2) | 0.40(2) | 0.72(2) | 11.2(4) | 10.2(3) |
| No MLIP Tool | **6.6(5)** | **6.8(4)** | **0.48(2)** | **0.46(2)** | 0.72(2) | **12.6(4)** | **11.1(3)** |
| No Generator Tools | 5.1(6) | 4.9(4) | 0.32(2) | 0.32(1) | 0.32(2) | 6.6(4) | 6.3(3) |
| No Structural Information | 5.0(4) | 6.5(4) | 0.40(1) | 0.41(1) | 0.75(2) | 11.1(4) | **11.1(3)** |
| *LLM Model* | | | | | | | |
| GPT 5.1 (default) | **5.4(5)** | **6.0(4)** | **0.40(2)** | **0.40(1)** | 0.71(3) | 10.6(4) | **10.4(3)** |
| Claude Sonnet 4.6 | 4.8(4) | 5.9(4) | 0.36(1) | 0.36(1) | 0.62(3) | **12.4(5)** | 8.5(3) |
| DeepSeek v3.2 | 3.6(3) | 3.5(2) | 0.27(1) | 0.26(1) | **0.82(2)** | 9.0(4) | 8.0(3) |
| GPT 5.4 Nano | 5.6(6) | 4.1(3) | 0.30(1) | 0.27(1) | 0.73(3) | 8.7(4) | 6.8(3) |

## C.6. Results on Additional Chemical Spaces

We also evaluate the generalization of discovery policies to different commercially relevant chemical families beyond the main results presented for intermetallics in Section 4. Table 6 shows results for discovery policies averaged across 10 systems in the chalcogenide, halide and oxide families respectively. The specific systems used are shown in Table 3.

We find that adaptive planning strategies significantly increase performance across all new families relative to the random baseline, mirroring the results in Table 1. Speedups vary by chemical family, with the random generation baseline performing particularly poorly for oxides, but relative gains remain consistent.

*Table 6.* Cross-system generalization results for discovery policies averaged across episodes, at a 0.1 eV stability threshold with a query budget of 50. **Higher** is better for all columns. The error in the final significant figure is given in brackets as the standard error in the mean. Details on metrics and experimental setup are given in Sections 3.3 and 4.

| Policy | | | Discovery Performance | | | | Discovery Diversity | | |
|---|---|---|---|---|---|---|---|---|---|
| Generator | Planner | Selector | AF | EF | AUDC | mSUN | Mean Comp. L1. | Unique Comps. | Unique SGs |
| *Chalcogenides* | | | | | | | | | |
| Random | Random | Random | 1.0(0) | 1.0(0) | 0.13(2) | 0.13(2) | 0.71(6) | 7.1(8) | 1.0(0) |
| Random | Diversity | Random | 1.8(4) | 1.3(1) | 0.12(1) | 0.13(1) | 0.78(6) | 6.5(6) | 1.0(0) |
| Random | LLM | Random | 1(1) | 0.6(2) | 0.024(5) | 0.022(4) | 0.27(5) | 1.8(2) | 1.0(0) |
| Chemeleon | Random | Random | 5(1) | 4.6(6) | 0.26(2) | 0.26(2) | 0.90(3) | 13.0(8) | 2.8(2) |
| Chemeleon | Diversity | Random | 3.4(5) | 4.3(5) | 0.23(1) | 0.24(1) | **0.97(3)** | 11.9(6) | 3.3(2) |
| Chemeleon | LLM | Random | 8(2) | 10(2) | 0.37(2) | 0.35(2) | 0.39(3) | 13.0(8) | 4.2(2) |
| Chemeleon | – | MLIP | 9(2) | **16(3)** | 0.41(5) | 0.41(5) | 0.58(4) | **20(2)** | 2.5(2) |
| LLM Orch. | – | – | **11(2)** | **16(3)** | **0.50(3)** | **0.49(3)** | 0.52(4) | 9.6(7) | **4.9(3)** |
| *Halides* | | | | | | | | | |
| Random | Random | Random | 1.0(0) | 1.0(0) | 0.39(2) | 0.39(2) | **1.00(1)** | 18.8(9) | 1.0(0) |
| Random | Diversity | Random | 0.55(8) | 0.98(2) | 0.36(1) | 0.37(1) | **1.00(1)** | 18.4(6) | 1.0(0) |
| Random | LLM | Random | 0.8(1) | 1.09(5) | 0.42(2) | 0.39(1) | 0.34(3) | 14.5(6) | 1.0(0) |
| Chemeleon | Random | Random | 1.34(9) | 1.42(5) | 0.55(3) | 0.54(2) | 0.98(1) | 27(1) | 3.0(2) |
| Chemeleon | Diversity | Random | 1.29(4) | 1.35(4) | 0.49(2) | 0.51(2) | 0.95(1) | 25(1) | 3.4(2) |
| Chemeleon | LLM | Random | 1.5(1) | 1.57(7) | 0.60(2) | 0.57(2) | 0.32(2) | 21.8(8) | 3.6(2) |
| Chemeleon | – | MLIP | 1.82(7) | 1.82(7) | 0.68(2) | 0.68(2) | 0.89(2) | **32(1)** | 3.5(2) |
| LLM Orch. | – | – | **2.4(1)** | **2.27(9)** | **0.83(1)** | **0.82(1)** | 0.41(4) | 12.4(6) | **5.3(3)** |
| *Oxides* | | | | | | | | | |
| Random | Random | Random | 1.0(0) | 1.0(0) | 0.046(6) | 0.050(6) | 0.75(9) | 3.1(3) | 1.0(0) |
| Random | Diversity | Random | 0.5(1) | 0.9(1) | 0.045(6) | 0.044(5) | 0.79(9) | 2.6(2) | 1.0(0) |
| Random | LLM | Random | 0.6(1) | 0.9(3) | 0.032(7) | 0.032(6) | 0.24(6) | 2.4(3) | 1.0(0) |
| Chemeleon | Random | Random | 3.0(4) | 3.4(5) | 0.12(1) | 0.12(1) | **0.86(6)** | 6.5(5) | 2.1(2) |
| Chemeleon | Diversity | Random | 1.2(1) | 2.0(2) | 0.066(6) | 0.068(5) | 0.79(7) | 3.5(2) | 1.8(1) |
| Chemeleon | LLM | Random | 12(2) | 16(3) | 0.33(2) | 0.30(2) | 0.27(1) | 12.2(7) | **3.8(3)** |
| Chemeleon | – | MLIP | **15(2)** | **22(4)** | **0.53(4)** | **0.50(4)** | 0.81(6) | **23(2)** | 3.4(2) |
| LLM Orch. | – | – | 11(2) | 13(1) | 0.42(3) | 0.40(3) | 0.24(2) | 8.3(7) | **3.8(3)** |

