# OpenReview forum: "MADE: Benchmark Environments for Closed-Loop Materials Discovery"
_ICML.cc/2026/Conference — ICML 2026 regular_

### Official Review · Reviewer_5eke · 2026-03-11

**Soundness:** 2
**Presentation:** 3
**Significance:** 2
**Originality:** 3
**Overall Recommendation:** 4
**Confidence:** 4

**Summary:**

In this work, the authors propose MADE, an environment for benchmarking the full materials discovery cycle. MADE decomposes the discovery process into Planner, Generator, Filter, and Selector, enabling users to freely combine different algorithms and analyse interactions among the modules. The authors also provide comprehensive experiments across a family of models to compare their performance.

**Compliance With Llm Reviewing Policy:**

Affirmed.

**Final Justification:**

The authors have addressed most of my concerns in the rebuttal. Therefore, I decide to raise my score to weak accept.

**Key Questions For Authors:**

Could the authors incorporate more material systems and methods in the rebuttal so that I can evaluate the scaling potential of the MADE benchmark?

In the limitation section, the authors mention that choosing MLIP as the Oracle is due to computational resource constraints. If the referee were changed from MLIP to DFT, would the change only affect search speed, or the entire search landscape?

**Limitations:**

yes

**Strengths And Weaknesses:**

**Strengths**

- Proposing an environment to evaluate the key components of material discovery is a nice try. Different from static predictive tasks, developing a framework for the entire lifecycle
- Within this framework, the users can delve into the interactions among the modules, which may provide more insights into the material discovery field.
- The presentation is good and makes the paper easy to understand.

**Weaknesses**
- The diversity of materials in MADE is limited. Specifically, the substances are metallic compounds in the MP. Oxides or semiconductors, which are extremely important in materials science, have not been explored in depth in the main experiments. The authors should at least provide some analysis of the behaviors of different materials systems.
- Another concern comes from the selection of the evaluation methodologies in the benchmark. Although the systems range from randomness to complex LLM-driven agents, the selected models are often introductory or representative of a particular field. For example, the diffusion-based generators (such as CDVAE) are not included in the experiment. Moreover, the performance of different LLMs and agents may also vary greatly. Testing only the GPT-5.1 and ReAct frameworks may not be representative enough.

---

> ### Author Rebuttal · Authors · 2026-03-31
>
> We thank the reviewer for describing the paper as *“easy to understand”* and noting that it may *"provide more insights into the material discovery field.”* MADE is designed as a **single, easily expandable framework** for benchmarking discovery. We demonstrate this by expanding the evaluation to include **3 additional materials families, 2 generators, and multiple LLM/agent frameworks**. Future expansions can easily be incorporated.
>
> ## Material Diversity
>
> > *"Oxides or semiconductors... have not been explored."*
>
> While we initially chose intermetallics as a complex discovery environment, we add **30 new chemical systems** (3-5 elements) across **Oxides, Halides, and Chalcogenides**, representing commercially critical spaces. The set up otherwise mirrors Table 1 in the paper.
>
> **Table 3: Performance on Diverse Chemical Systems**
> | System | Strategy | AF | EF | AUDC | mSUN |
> |---|---|---|---|---|---|
> | Chalcogenides, e.g. Cu-Zn-Sn-S | Chemeleon | 5(1) | 4.6(6) | 0.26(2) | 0.26(2) |
> |  | Chemeleon + Diversity | 3.4(5) | 4.3(5) | 0.23(1) | 0.24(1) |
> |  | Chemeleon + LLM | 8(2) | 10(2) | 0.37(2) | 0.35(2) |
> |  | Chemeleon + MLIP | 9(2) | 16(3) | 0.41(5) | 0.41(5) |
> |  | LLM Orchestrator | **11(2)** | **16(3)** | **0.50(3)** | **0.49(3)** |
> | Halides, e.g. Li-In-Cl | Chemeleon | 1.34(9) | 1.42(5) | 0.55(3) | 0.54(2) |
> |  | Chemeleon + Diversity | 1.29(4) | 1.35(4) | 0.49(2) | 0.51(2) |
> |  | Chemeleon + LLM | 1.5(1) | 1.57(7) | 0.60(2) | 0.57(2) |
> |  | Chemeleon + MLIP | 1.82(7) | 1.82(7) | 0.68(2) | 0.68(2) |
> |  | LLM Orchestrator | **2.4(1)** | **2.27(9)** | **0.83(1)** | **0.82(1)** |
> | Oxides, e.g. Na-Fe-Mn-Ni-O | Chemeleon | 3.0(4) | 3.4(5) | 0.12(1) | 0.12(1) |
> |  | Chemeleon + Diversity | 1.2(1) | 2.0(2) | 0.066(6) | 0.068(5) |
> |  | Chemeleon + LLM | 12(2) | 16(3) | 0.33(2) | 0.30(2) |
> |  | Chemeleon + MLIP | **15(2)** | **22(4)** | **0.53(4)** | **0.50(4)** |
> |  | LLM Orchestrator | 11(2) | 13(1) | 0.42(3) | 0.40(3) |
>
> **Key trends:** Planning significantly increases performance across all new families relative to the random baseline, mirroring intermetallics. Speedups vary by system (halides/intermetallics are harder search spaces), but relative gains remain consistent.
>
> ## Selection of Baselines and LLMs
>
> > *"diffusion-based generators… are not included... GPT-5.1 and ReAct ... may not be representative."*
>
> MADE's modularity allowed quick integration of additional models. Beyond Chemeleon [1], we add **MatterGen** [2] and **OMatG** [3], which are SOTA on LeMat-GenBench [4].
>
>
> **Table 4: Additional Generative Model Baselines on Intermetallics**
> | Strategy | AF | EF | AUDC | mSUN |
> |---|---|---|---|---|
> | Chemeleon | 1.7(1) | 1.70(9) | 0.19(1) | 0.19(1) |
> | Chemeleon + Diversity | 2.1(2) | 2.0(1) | 0.19(2) | 0.21(2) |
> | Chemeleon + LLM | **3.9(4)** | **3.3(2)** | **0.27(2)** | **0.26(2)** |
> | OMatG | 1.8(2) | 1.71(8) | 0.19(1) | 0.18(1) |
> | OMatG + Diversity | 1.5(1) | 1.8(1) | 0.16(2) | 0.17(2) |
> | *OMatG + LLM | **3.7(6)** | 3.0(2) | 0.19(1) | 0.19(1) |
> | *MatterGen | 2.5(4) | 2.3(3) | **0.28(3)** | **0.25(3)** |
>
> *results for 25/30 systems (ongoing runs). Note: non-MP-20 models (MatterGen/OmatG) may have train/test overlap.
>
> We show LLM/agent ablations in **Tables 1,2** (response to Reviewer Ek6N). Overall, these additional results show that **agentic and systematic planning consistently outperform isolated generation** across chemical spaces and generators.
>
> ## MLIP vs. DFT Oracles
>
> >If the referee were changed from MLIP to DFT, would the change only affect search speed, or the entire search landscape?
>
> Changing the oracle to DFT affects the **landscape precision**. Modern MLIPs are trained on a mix of DFT and DFT+U data; the latter is essential for transition metal oxides. Consequently, MLIP potentials reflect a **statistical average** of these settings. While DFT would resolve fine-grained energy differences, the relative topology is largely consistent. We note that MLIPs performance can degrade out of distribution compared to DFT [5], so this may affect the landscape in new chemical spaces. We investigate similar phenomena in Sec. 3.4.
>
> Crucially, MADE functions as a **"pre-deployment" simulator**. MLIP proxies allow scalable in-silico discovery strategy optimization before committing to expensive DFT/lab resources.
>
> **In summary**: We have (1) expanded evaluation to 60 diverse chemical systems, (2) integrated current SOTA generative models and LLMs, and (3) clarified the MLIP to DFT gap. We hope these results confirm MADE's utility as a rigorous, generalizable benchmark.
>
> [1] [Park et al 2025](https://doi.org/10.1038/s41467-025-59636-y)
>
> [2] [Zeni et al 2025](https://doi.org/10.1038/s41586-025-08628-5)
>
> [3] Höllmer et al. "Open Materials Generation with Stochastic Interpolants." International Conference on Machine Learning. PMLR, 2025.
>
> [4] [LeMat-GenBench leaderboard](https://huggingface.co/spaces/LeMaterial/LeMat-GenBench)
>
> [5] [Coscia et al 2025](https://doi.org/10.48550/arXiv.2508.14022)

---

> > ### Author Rebuttal · Reviewer_5eke · 2026-04-04
> >
> > The authors' rebuttal has resolved most of my concerns.

---

> > > ### Author Response · Authors · 2026-04-04
> > >
> > > We are pleased that our rebuttal and additional experiments have resolved most of your concerns - your feedback has strengthened the paper significantly.
> > >
> > > We kindly ask that you consider raising your score to reflect this updated assessment of our work.

---

### Official Review · Reviewer_Ek6N · 2026-03-12

**Soundness:** 3
**Presentation:** 3
**Significance:** 3
**Originality:** 2
**Overall Recommendation:** 4
**Confidence:** 4

**Summary:**

This paper introduces a benchmark framework intended to evaluate end-to-end, closed-loop computational materials discovery pipelines under a limited oracle query budget. The environment formalizes discovery as sequential proposal/evaluation of crystal structures, with success defined as finding novel, unique, (meta)stable materials relative to a convex hull. Methods are evaluated using both independent metrics (e.g. mSUN, AUDC) and relative metrics (Acceleration Factor, Enhancement Factor) against a baseline policy.

**Compliance With Llm Reviewing Policy:**

Affirmed.

**Key Questions For Authors:**

1. The paper assumes oracle cost dominates, but some policies use different levels of compute (e.g. 1024-generation + MLIP ranking). How sensitive are conclusions if intermediate compute is also counted?
2. Since initial hull materials, generator training, and (some) surrogate tooling are trained on the same dataset, how robust are AF/EF improvements when the generator is out-of-distribution relative to the oracle or H0 is drawn from a different dataset?
3. For the LLM orchestrator, which components matter most? Did you try different tool sets, buffer/history, ReAct iteration cap, or access to MLIP scoring?

**Limitations:**

yes

**Strengths And Weaknesses:**

### Strengths
1. The paper clearly motivates why static predictive/generative benchmarks miss the sequential nature of closed-loop discovery, and defines appropriate metrics (mSUN, AUDC, AF/EF) to evaluate it.
2. The modularity of the planner/generator/filter/selector plus an agentic orchestrator makes it easy to ablate gains for each component (e.g. MLIP selection vs LLM planning) and to extend the benchmark later.
3. The results include useful empirical observations how planning’s value grows with system size and how surrogate ranking degrades at stricter stability tolerances.

### Weaknesses
1. The framework argues that oracle evaluations dominate cost (motivating query-budget metrics), but the reported experiments use a cheap MLIP oracle rather than DFT/experiment. This is reasonable for scale, but means the benchmark’s setting of an expensive oracle is not demonstrated in the main results.
2. Potential coupling / shared bias across components. The experiments rely on generators and models trained on MP-20 (the initial set also comes from MP-20 via Materials Project). The paper acknowledges this as a limitation.
3. Some policies differ in how they generate candidates sequentially vs in parallel, which is not quite a fair comparison, if time/compute per generation (not just oracle calls) becomes a bottleneck in practice.

---

> ### Author Rebuttal · Authors · 2026-03-31
>
> We thank the reviewer for noting that our paper *"clearly motivates why static predictive/generative benchmarks miss the sequential nature of closed-loop discovery"*. We address the specific concerns below:
>
> ## Oracle Cost and Intermediate Compute
>
> > *"...cheap MLIP oracle rather than DFT…” “policies differ in how they generate candidates…” “how sensitive are conclusions if intermediate compute is also counted?"*
>
> We agree that DFT/experimental oracles are the ultimate goal. MADE **emulates** this setting. In our experiments, We use high-fidelity MLIP as a **scalable proxy** to compare discovery algorithms efficiently:
> - **We assume oracle costs dominate**: Real-world oracles (DFT/wet-lab) are costlier by orders of magnitude than ML inference; thus, strategy-specific intermediary compute is negligible.
> - **Pre-Deployment Simulator**: MADE is designed as a **modular simulator** where discovery agents can be tuned **in-silico** before expensive deployment. MLIP proxies enable **efficient** testing of acceleration.
> - **Scaling Laws**: While we normalized candidate generations to maintain fairness comparing generative models (response to Reviewer 5eke), MADE uniquely enables the **study of scaling laws for test-time compute**. E.g. identifying thresholds where additional planning/generations meets diminishing returns relative to discovery speed is a compelling research direction.
>
> Therefore, we argue that conclusions regarding the **relative ranking** of discovery policies are robust when counting intermediate compute. While agentic planners and MLIP rankings have higher intermediate costs than random search, they can achieve **6x reduction in required oracle queries** (Table 1 in the paper), which is the **primary bottleneck** in practice.
>
> ## Bias and Out-of-Distribution (OOD) Robustness
>
> > *"potential coupling…"* *“how robust are AF/EF improvements… out-of-distribution?”*
>
> We recognize the reviewer's concern regarding training distribution (MP-20) coupling between the generator, surrogate, and $H_0$. We address this by highlighting the following:
> - **Discovery is by definition OOD**: The objective is to find stable materials not in $H_0$. Our framework is designed to evaluate the agent's ability to transition from known to unknown chemical spaces.
> - **Privileged Oracle**:  We align $H_0$ and generators (MP-20) for controlled OOD testing while using a more "privileged" oracle (trained also on Alexandria/OMAT) than agent tools to simulate higher-fidelity, OOD experimental feedback.
> - **Empirical Robustness**: New results on Oxides, Halides, and Chalcogenides (see response to Reviewer Ek6N) confirm that discovery gains persist across diverse chemical families.
> - **Diagnostic Utility**: MADE serves as a diagnostic environment for these specific concerns. Section 3.4 shows that performance degrades when tasks require sensitivity beyond surrogate limits, capturing sim-to-real gaps.
>
> In summary, MADE provides the modular infrastructure that allows users to **measure how specific OOD settings impact discovery rates**, giving another interesting direction for future controlled studies.
>
> ## Agent Ablation
>
> > *"For the LLM orchestrator, which components matter most?"*
>
> We conducted an extensive ablation study across agent components and model providers (same set up as in Table 1 in paper) to isolate these effects.
>
> **Table 1: Agent Ablation**
> | LLM Orch. Ablation | AF | EF | AUDC | mSUN |
> |---|---|---|---|---|
> | Original (ReAct k=10, history k=20, MLIP, structural info, chemeleon generator) | 5.4(5) | 6.0(4) | 0.40(2) | 0.40(1) |
> | ReAct (k=5) | 5.4(4) | 6.1(4) | 0.41(2) | 0.40(1) |
> | ReAct (k=20) | 5.9(6) | 6.0(4) | 0.40(2) | 0.40(1) |
> | History (k=5) | 5.2(5) | 6.0(4) | 0.40(2) | 0.40(2) |
> | No MLIP | **6.6(5)** | **6.8(4)** | **0.48(2)** | **0.46(2)** |
> | No Generator | 5.1(6) | 4.9(4) | 0.32(2) | 0.32(1) |
> | No Structural Info | 5.0(4) | 6.5(4) | 0.40(1) | 0.41(1) |
>
> **Table 2: Model Comparison**
> | Orch. Model | AF | EF | AUDC | mSUN |
> |---|---|---|---|---|
> | GPT 5.1 | **5.4(5)**| **6.0(4)**| **0.40(2)** | **0.40(1)**|
> | Sonnet 4.6 | 4.8(4) | **5.9(4)**| 0.36(1) | 0.36(1) |
> | DeepSeek v3.2 | 3.6(3) | 3.5(2) | 0.27(1) | 0.26(1) |
> | GPT 5.4 Nano | **5.6(6)** | 4.1(3) | 0.30(1) | 0.27(1) |
>
> Key conclusions include:
> - **Tooling**: Removing the MLIP scorer (**No MLIP**) actually increases performance, meaning that agentic planning can be more effective than surrogate ranking that could lead the agent astray. Interestingly, only small reductions in performance are observed in the absence of generative tools (**No Generator**). LLM proposed structures are reasonable.
> - **Reasoning**: Performance scales with reasoning depth (**ReAct k steps**), and not as significantly with memory (**History k structures**).
> - **Structural Context**: Performance degrades without structural information.
> - **Models**: Gains vary across model sizes/costs (GPT 5.4 Nano) and providers (Claude, DeepSeek), but all show speed ups.

---

> > ### Author Rebuttal · Reviewer_Ek6N · 2026-04-02
> >
> > I thank the authors for their reply.
> > Overall, I am doubtful if the benchmark will see meaningful adaption, and do not see key clear insights from the experiments in between the many smaller "anecdotal" findings (that removing the MLIP scorer increases performance in some cases, without further investigation, causes more confusion than it helps here). I keep my score.

---

> > > ### Author Response · Authors · 2026-04-04
> > >
> > > We thank the reviewer for their continued engagement. We want to clarify that our **primary contribution** is providing the **modular and extensible infrastructure and framework** required to rigorously study and advance autonomous materials discovery (rather than any single empirical insight). As the first end-to-end framework for evaluating discovery under constrained oracle budgets (to our knowledge), MADE provides a realistic pre-deployment simulation using grounded discovery metrics that existing static benchmarks cannot capture. We believe this extensible framework provides the necessary platform for the community to build upon to explore and optimize real-world discovery campaigns.

---

### Official Review · Reviewer_cj6D · 2026-03-13

**Soundness:** 3
**Presentation:** 3
**Significance:** 3
**Originality:** 3
**Overall Recommendation:** 5
**Confidence:** 2

**Summary:**

This paper presents MADE, a benchmarking framework for dynamic, trial-and-error-based discovery in material sciences. In a domain constrained by real-world laboratory evaluation steps, ambitions to automate materials science research must show efficient dynamic updating and planning in the face of experimental evidence. To address this, this paper develops a benchmark for measuring autonomous material science systems’ efficiency in terms of discoveries per evaluation (reported as ‘acceleration’ and ‘enhancement’ factor). Using several semi-random strategies as a baseline, it utilizes the benchmark to quantify discovery acceleration and enhancement relative to baseline for several discovery policies in a modular way. Specifically, it compares different combinations of random strategies, LLMs, MLIPs, Chemeleon, and Euclidean diversity for planning, generating, filtering, and selecting materials during the discovery loop. Material evaluation after each selection is simulated via a higher-fidelity MLIP.

**Compliance With Llm Reviewing Policy:**

Affirmed.

**Key Questions For Authors:**

n/a

**Limitations:**

yes

**Strengths And Weaknesses:**

The paper appears scientifically and technically sound, with claims appearing well-supported and experiments well-designed. The problem of measuring dynamic discovery is presented and formalized well, and the system to benchmark around it seems well-thought-out and fit for testing advances on that problem. The paper and core contributions are presented well, it is well-structured, and written clearly. Its figures underline the overall contributions and claims well. The paper appears to make significant contributions to the development and testing of automated material discovery systems in material sciences. This work appears to present the first dynamic benchmark of material science discovery pipelines, constituting a novel contribution.

---

> ### Author Rebuttal · Authors · 2026-03-31
>
> We thank the reviewer for their strong support, noting that the the system is *"well-thought-out"* and that MADE is *"a novel contribution"* as the *"first dynamic benchmark of material science discovery pipelines."*.
>
> To further reinforce the reviewer's point that our framework is *"fit for testing advances on that problem,"*, we have substantially expanded our evaluation suite to demonstrate the framework's robustness across different generative models, LLMs, agent configurations and chemical systems (see response to Reviewers 5eke and Ek6N).

---

> > ### Author Rebuttal · Reviewer_cj6D · 2026-04-05
> >
> > Great to see.

---

### Official Review · Reviewer_crX5 · 2026-03-13

**Soundness:** 3
**Presentation:** 2
**Significance:** 3
**Originality:** 3
**Overall Recommendation:** 4
**Confidence:** 3

**Summary:**

The authors introduce MADE, a framework for comparing materials discovery pipelines.  Materials discovery is a growing field with many opportunities for automation and AI.  The authors propose a method for evaluating materials discovery techniques and evaluate them on the Materials Project.  They report that systems using surrogate models perform better for simple systems, while techniques that involve adaptation are needed for more complex systems.

**Compliance With Llm Reviewing Policy:**

Affirmed.

**Final Justification:**

I have changed my recommendation as a result of the author rebuttal, which addressed my concerns about presentation.

**Key Questions For Authors:**

[No questions]

**Limitations:**

Yes.

**Strengths And Weaknesses:**

The paper appears technically sound and is well-written.  The experiments support the paper's conclusions, although I believe that the final results could be more clearly stated in the abstract and that more detail about the experiments could be included in the main body of the paper.  The paper seeks to remain general, but more grounding in specific techniques (for example, choosing a specific "oracle" approach) might strengthen the work.  The work is original, but the impact could be stronger.  A drawback of the current presentation is the overall flow and figures.  The placement of the related work at the end of the paper seems an unusual choice, as does the current position of Figure 1.  Additionally, the text on Figure 7 is too small to read in its current form.  This paper employs sound techniques and good reasoning, but the presentation should be addressed.

---

> ### Author Rebuttal · Authors · 2026-03-31
>
> We thank the reviewer for the constructive feedback and for noting that the paper is *"technically sound and is well-written"* and employs *"sound techniques and good reasoning."* We appreciate the specific guidance on presentation and grounding which we address below and will update the paper accordingly in the revision.
>
> ## Presentation and Flow
>
> We acknowledge the reviewer's points regarding layout and clarity. We are committed to the following structural updates for the final version:
> - **Abstract**: We will explicitly state the core takeaways (e.g., planning/agentic gains scale with chemical complexity; surrogate models excel in simpler regimes).
> - **Layout & Figures**: We will move "Related Work" to Section 2. All figures, specifically **Figure 7**, will be updated with increased font sizes for legibility, and we will address the placement of the figures to match the flow of the paper.
> - **Experiment Detail**: Key implementation details currently in the Appendix will be moved to Section 3 to ensure the main body is self-contained.
>
> ## Grounding in Specific Techniques
>
> > *“...more grounding in specific techniques (e.g., choosing a specific oracle) might strengthen the work.”*
>
> We agree that concrete grounding strengthens the framework's utility. While MADE is modular by design, **our experiments do instantiate a specific, realistic discovery setting** where an **expensive oracle** acts as the bottleneck (see also: response to Reviewer Ek6N). While we use high-fidelity MLIPs [1] for scalable benchmarking, and fixed $H_0$ (the Materials Project catalog), the framework treats the oracle as a **'black box’**, making it well suited for grounding in expensive DFT or wet-lab setups which are widely prevalent in real-world materials discovery.
>
> We show that MADE provides the necessary framework to compare the performance of discovery algorithms **in silico, before deployment** within an end-to-end discovery loop. Our framework remains **flexible** to swap these for expensive oracles for real-world deployment, change $H_0$ with new datasets, or change generators and agents when new models or techniques come out. We showcase this by substantially expanding the evaluation suite (response to Reviewers 5eke and Ek6N).
>
> We hope these commitments to revising the presentation of the paper and additional clarifications alleviate your concerns. We remain available for any further clarifications to help improve your assessment of the work.
>
> [1] [Rhodes et al 2025](https://doi.org/10.48550/arXiv.2504.06231)

---

> > ### Author Rebuttal · Reviewer_crX5 · 2026-04-04
> >
> > I appreciate the authors' response to my comments about presentation and layout.  I have changed my score assuming that the changes described in the rebuttal are incorporated into the final version.

---

### Decision · Program_Chairs · 2026-04-30

**Decision:**

Accept (regular)

**Comment:**

This paper has received four reviews and the discussion with some reviewers has been unfortunately rather limited. While no reviewer has expressed outstanding enthusiasm about the paper and they have expressed doubts about the potential impact, the reviewers have indicated that the paper is technically sound and no serious concerns about the quality have remained after the rebuttal. My own assessment of the paper is in line with some reviews, in the sense that I have high uncertainty about whether this benchmark will be widely adopted by the materials discovery community. At the same time, I think the paper is technically correct, well written and it should be useful to part of the ICML community. Therefore, to follow the ICML criteria, my recommendation is to accept this paper for presentation.